# A Role of Exopolysaccharide Produced by *Streptococcus thermophilus* in the Intestinal Inflammation and Mucosal Barrier in Caco-2 Monolayer and Dextran Sulphate Sodium-Induced Experimental Murine Colitis

**DOI:** 10.3390/molecules24030513

**Published:** 2019-01-31

**Authors:** Yun Chen, Ming Zhang, Fazheng Ren

**Affiliations:** 1Key Laboratory of Functional Dairy, College of Food Science and Nutritional Engineering, China Agricultural University, Beijing 100083, China; chenyun2018MN@163.com (Y.C.); renfz2018fdl@163.com (F.R.); 2School of Food and Chemical Engineering, Beijing Technology and Business University, Beijing 100048, China

**Keywords:** *Streptococcus thermophiles*, exopolysaccharide, DSS-induced colitis, tight junction protein

## Abstract

Exopolysaccharide (EPS) produced by probiotics may play an important role in gastrointestinal disease prevention, including ulcerative colitis. However, there is no literature reporting on the intervention effects of purified EPS. The aim of this study was to investigate the alleviating effect of the purified EPS produced by *Streptococcus thermophilus* MN-BM-A01 on murine model of colitis induced by dextran sulphate sodium (DSS). A water-soluble heteropolysaccharide (EPS-1) isolated from MN-BM-A01 was composed of rhamnose, glucose, galactose, and mannose in a molar ratio of 12.9:26.0:60.9:0.25, with molecular weight of 4.23 × 10^5^ Da. After EPS-1 administration, the disease severity of mouse colitis was significantly alleviated, mainly manifesting as the decrease of disease activity index and mitigated colonic epithelial cell injury. Meanwhile, pro-inflammatory cytokines levels (tumor necrosis factor-α, interleukin-6, and interferon-γ) were significantly suppressed, the reduced expressions of tight junction protein (claudin-1, occludin, and E-canherin) were counteracted. In addition, the results in vitro showed that EPS-1 protected intestinal barrier integrity from the disruption by lipopolysaccharide in Caco-2 monolayer, increased expression of tight junction and alleviated pro-inflammatory response. Collectively, our study confirmed the protective effects of purified EPS produced by *Streptococcus thermophilus* on acute colitis via alleviating intestinal inflammation and improving mucosal barrier function.

## 1. Introduction

Ulcerative colitis (UC) is a set of complicated chronic inflammatory and ulceration conditions of the colonic mucosa, accompanied by clinical symptoms such as diarrhea, rectal bleeding, body weight loss, and abdominal pain [1]. According to clinical observation, UC is easy to relapse and difficult to permanently cure. Epidemiological data revealed that the incidence of UC has been significantly increasing over the past two decades [2]. A lack of effective strategy was the most important cause of high morbidity. Therefore, finding a proper treatment is necessary.

While the precise mechanism of UC is still unclear, there is no doubt that intestinal mucosal barrier dysregulation and increased paracellular permeability play a critical role in the pathogenesis of UC [3]. Tight junctions (TJ) of enterocyte regulates the integrity of intestinal barrier predominantly, which influence the paracellular and transcellular transport [4]. Various transmembrane proteins including claudins, occludin and Zonula Occludens-1 (ZO-1) often considered to regulate intestinal permeability [5]. Many researches pointed out that, in UC patients, downregulations of tight junctions giving rise to the invasion of the colon wall with intestinal pathogens and toxins, which was considered as a vital event in the pathogenesis of intestinal inflammation [6,7,8]. While the molecular mechanisms that mediate the decreased expression of TJ proteins remain poorly characterized, increased production of pro-inflammatory cytokines, such as tumor necrosis factor (TNF)-α and interferon (IFN)-γ was considered as the major upstream event leading to the disruption of the gut barrier [5,9]. These cytokines could alter the structure of the TJ by inducing the expression and activity of the Myosin Light Chain Kinase and/or triggering the endocytosis of TJ proteins [10]. In addition, anti-inflammatory cytokines (marked by upregulation of interleukin (IL)-13, IL-4, and IL-10) have been shown to be directly involved in the epithelial barrier function [9,11]. The emerging novel biological therapies targeting pro-inflammatory and anti-inflammatory cytokines balance were also reported [12,13].

Recently, increasingly evidences supported that probiotic supply beneficial effects for host health, especially in terms of intestinal function improvement and prevention of a variety of intestinal diseases including UC [14,15]. In these reports, exopolysaccharides (EPSs) was often considered as one of the proposed mechanisms mediating (some of) these health benefits, because they aid in adherence and colonization within the human host [16,17]. On this basis, EPS also may act as intermediaries in the cross-talk between mammalian hosts and probiotic bacteria, interacted with receptors located in the gut epithelium [18], and are involved in the regulation of the host immune system [19]. Neriman Sengul et al. have showed that EPS-producing probiotic bacteria significantly attenuate experimental colitis by anti-oxidative stress [20]. Claudio Hidalgo et al. also showed that EPS-producing strains of *B. animalis subsp. lactis* could be a good candidate to check its anti-inflammatory ability in patients suffering from intestinal inflammation [21]. However, there is no literature reporting on the intervention effects of purified EPS in experimental colitis.

MN-BM-A01 (CGMCC No. 11383) was is a *Streptococcus thermophiles* (*S. thermophilus*) strain isolated from Yogurt Block in Gansu, China. *S. thermophilus* MN-BM-A01 could produce a high level of EPS, which can confer the yogurt with improved rheological properties, the maximum yield of EPS produced by MN-BM-A01 strains could reach 20.50 mg/L. The genomic sequence indicated that this strain included a 35.3-kb gene cluster involved in EPS biosynthesis [22]. However, its biological function of EPS from this strain was unclear. The aim of this study was to investigate the alleviating effect and the possible mechanisms of the purified EPS on the murine model of colitis induced by dextran sulphate sodium (DSS).

## 2. Results

### 2.1. The Molecular Mass and Monosaccharide Composition of the EPS

The crude EPS from the culture supernatant of MN-BM-A01 was first prepared by protein removal and ethanol precipitation. By anion-exchange chromatography of DEAE Sepharose Fast flow, the crude EPS was separated into three main fractions, namely EPS-1, EPS-2, and EPS-3. A fraction profile was shown in Figure 1A. Sub-fraction EPS-1 was the main component, which classified as neutral polysaccharides according to its soluble characteristics.

The molecular mass of the EPS-1 was determined by gel-permeation chromatography (GPC, Figure 1B). The chromatogram of the EPS-1 appeared as a single symmetrical narrow peak, confirming the homogeneity of the purified EPS sample. The molecular mass was calculated as 423168.7 (4.23 × 10^5^) Da, according to the standard curve equation Log Mw = −0.1741x + 11.505 (R^2^ = 0.9913), where Mw is the peak molecular weight and x is the retention time. GC-MS analysis of the monosaccharide composition of the EPS-1 showed that the EPS was composed of different sugar monomers including rhamnose, glucose, galactose, and mannose in an approximate molar ratio of 12.9:26.0:60.9:0.25 (Figure 1C, Table 1), suggesting that the EPS-1 was a heteropolysaccharide.

### 2.2. EPS-1 Alleviated the Clinical Symptoms of DSS-Induced Colitis in Mice

There were no significant differences in body weight between the treatment groups at the beginning of the experiment (*p* > 0.05). When mice were treated with DSS for seven consecutive days, body weight was reduced by 11.6% compared to the control group (Figure 2A). Colon length is an important indicator of the incidence of colitis. DSS treatment shortened colon length by 23.3% (*p* < 0.05). EPS-1 (200 mg/kg) significantly alleviated the effects of DSS on body weight loss and colon shortening (Figure 2A,B). 

The grade of ulcerative colitis induced by DSS was evaluated by the disease activity index (DAI) score, which was the sum of scores given for body weight loss, stool consistency, and presence of fecal blood. DAI scores in the four groups of mice are shown in Figure 2C. A significant increase of DAI score was observed in the DSS-treated group compared with the control groups (*p* < 0.05). In two EPS-1 treatment groups, the DAI scores were significantly decreased when compared to the DSS group (*p* < 0.05), indicating that EPS-1 could significantly alleviate the clinical symptoms of DSS-induced colitis in mice.

Histologic examination of the colon revealed epithelial injury and the degree of inflammation. The colons from all of the mice in each group were examined in hematoxylin-eosin (HE) stained slides. According to Figure 2E, DSS-only treated mice displayed the most severe infiltration of inflammatory cells, disruption of surface epithelium, and loss of crypts. Oral EPS-1 administration groups showed less severe colitis compared to the DSS-only treated group, but they were more severe when compared to the control group. Histological scores in the four groups of mice are shown in Figure 2D. At the end of the experiment (day 14), histological scores were determined. The histological scores in DSS-treated group increased compared to the control group. When EPS-1 was given for the whole experimental period, the decrease of the histological scores were remarkable, while it was still higher when compared to the control group (*p* < 0.05).

### 2.3. EPS-1 Promoted a Change in Pro-Inflammatory and Anti-Inflammatory Cytokines 

The DSS-induced colitis was accompanied by the imbalance of pro-inflammatory and anti-inflammatory cytokines, which was usually regarded as major suspects in the formation of colitis. To understand the mechanism that underlies the alleviation of DSS-induced colitis in mice after treatment with EPS-1, we examined the levels of Th1 pro-inflammatory (TNF-α, IL-6, and IFN-γ) and anti-inflammatory (IL-4 and IL-10) in colon tissues. As shown in Figure 3A, in DSS treatment group, the expression of the IFN-γ, IL-6 and TNF-α in colon tissues was significantly increased (*p* < 0.05). When EPS-1 was given for the whole experimental period, a remarkable decrease of the pro-inflammatory cytokine secretion (IFN-γ, IL-6, and TNF-α) was observed (*p* < 0.05). In addition, the expression of anti-inflammatory cytokines were also detected in the colon tissues of all mice. As shown in Figure 3B, although IL-4 and IL-10 levels varied slightly after treatment with DSS and/or EPS-1, the difference was not significant when compared with the control group (*p* < 0.05).

### 2.4. EPS-1 Relieved DSS-Induced Colonic Epithelial Tight Junction Disruption in Mice 

Tight junction (TJ) is an essential permeable intercellular barrier and plays a critical role in the intestinal epithelial barrier integrity. As shown in Figure 4, we observed that the expressions of tight junction proteins, such as claudin-1, occludin, and E-cadherin remarkably reduced in the colonic tissues of DSS induced colitis mice by Western-Blot. After EPS-1 administration, the reduced expressions of claudin-1, occludin, and E-canherin proteins were counteracted (Figure 4A,B), which provided another powerful evidence for the protective effects of EPS-1 on mice colitis.

### 2.5. EPS-1 Protected against Colonic Epithelial Tight Junction Disruption In Vitro

In order to further explicit the protective effects of EPS-1 on acute colitis, the continuous line of human epithelial colorectal adenocarcinoma, Caco-2 cell line was employed to carry out the in vitro experiment. Firstly, we tested the transepithelial electrical resistance (TER) values of Caco-2 monolayers treated with lipopolysaccharide (LPS, 10 μg/mL) without or with EPS-1 (2 mg/mL) and EPS-1 (2 mg/mL) alone for 24 h. Compared with the no treatment cells (Control), as shown in Figure 5A, LPS caused a remarkable decrease of the TER levels after 24 h. By contrast, EPS-1 significantly dampened the LPS-induced drop of TER. Fluorescein isothiocyanate (FITC)-dextran measurement in the Caco-2 monolayers showed 77.8% increase in the intensity of 4 kDa FITC-dextran in the lower chamber of the Transwell cell cultures compared to the controls (*p* < 0.05). This improvement was counteracted significantly to 120.5 ± 21.9% of the control by treatment with EPS-1 (*p* < 0.05; Figure 5B). Consistent with the changes of TER and FITC-dextran measurement, as presented in Figure 5D,E, the expression of E-cadherin, occludin, and claudin-1 decreased in LPS group. EPS-1 remarkably alleviated the morphological disturbances of E-cadherin and claudin-1. These results suggest that EPS-1 protect intestinal barrier function from the disruption by LPS in Caco-2 monolayers.

To confirm if EPS-1 has an effect on inflammatory immune responses, we tested the levels of pro-inflammatory cytokines secreted by Caco-2 monolayers treated with LPS without or with EPS-1 for 24 h. As demonstrated in Figure 5C, the levels of pro-inflammatory cytokines (TNF-α, IFN-γ, and IL-6) significantly increased after treatment of LPS, a typical inflammatory activator. In contrast, consistent with the results of animal experiments, EPS-1 diminished the LPS-induced increase of pro-inflammatory cytokines (*p* < 0.05).

## 3. Discussion

Ulcerative colitis (UC) is a set of complicated chronic inflammatory and ulceration conditions of the colonic mucosa, accompanied by clinical symptoms [23]. Recently, increasingly, evidences supported that probiotics conferred a lot of health benefits, including UC prevention [15]. EPS may play a role in this prevention process [16]. However, there is no literature reporting on the intervention effects of purified EPS in experimental colitis. In this study, we verified that exopolysaccharide produced by *S. thermophilus* could alleviate the dextran sulfate sodium-induced experimental colitis in mice. The reduced severity of the disease after EPS treatment was associated with an inhibition of pro-inflammatory cytokine in the lymphocytes of colon. Furthermore, we identified that intestinal barrier function was significantly ameliorated in mice and Caco-2 monolayers after EPS treatment compared to the non-treated control mice.

*S. thermophilus* is the most important dairy starter. The exopolysaccharides of *S. thermophilus* can improve the properties of the dairy product and confer benefcial health effects [24]. Some studies indicated that EPS structure is very diverse, and has a close relationship with its functions [25,26]. Most *S. thermophilus* EPSs are heteropolysaccharide [27], which were composed of repeating subunits of three–eight different monosaccharides, and were predominantly composed of galactose, glucose, rhamnose, and *N*-acetyl-galactosamine in different ratios [28]. In this study, the EPS-1 produced by *S. thermophilus* MN-BM-A01 was a representative heteropolysaccharide, consisting of rhamnose, glucose, galactose, and mannose in an approximate molar ratio of 12.9:26.0:60.9:0.25.

The occurrence and development of UC were strongly associated with the imbalance of pro-inflammatory and anti-inflammatory cytokines [12]. A previous report showed that a UC-like mouse model of DSS-induced colitis had neutrophil accumulation and increased expression of pro-inflammatory in the colon [29]. TNF-α, a very potent matrix metalloproteinases produced by neutrophils and activated macrophages contribute to the epithelial ulceration and sub-mucosal destruction [30]. Progressive release of cytokines IFN-γ and IL-6 from inflamed colon, produced by T cells and macrophages was also shown to be correlated with development of colitis. Administration of a neutralization antibody against IFN-γ significantly ameliorated colonic inflammation in DSS-induced colitis in mice, and IFN-γ-/-mice [31,32]. Taken together, the release of pro-inflammatory cytokines in this acute colitis model is the result of rapid recruitment and activity of macrophages and neutrophils and initiating lymphocyte proliferation and activity [33,34]. Our study also demonstrated that production of pro-inflammatory cytokines increased significantly, but anti-inflammatory cytokines (IL-4 and IL-10) varied slightly in UC mice. 

Inhibition of pro-inflammatory cytokines has been used as a major strategy to attenuate colitis. Although therapeutic effects of probiotics on colitis were strain-specific and dose-dependant, the anti-inflammatory effects of probiotics play a decisive role [35,36]. A previous study has indicated that the expression of TNF-α, IL-1β, and IL-6 in the colon tissues was diminished dose-dependently by a mixture of three potential probiotic strains (*Lactobacillus johnsonii* IDCC9203, *Lactobacillus plantarum* IDCC3501, and *Bifidobacterium animalis subspecies lactis* IDCC4301) in DSS induced colitis [37]. Pan, T. et al. also illustrated that an appropriate dose of *Lactobacillus paracasei subsp. paracasei* LC-01 can prevent intestinal damage in mice with DSS-induced colitis by inhibiting the expression of inflammatory cytokines [38]. In our study, EPS-1 treatment attenuated the release of TNF-α, IFN-γ and IL-6 significantly, suggesting that this effect is the primary cause of anti-colitic activity of EPS. The inhibition effect or pro-inflammatory cytokines was further confirmed in Caco-2 cell lines treated by LPS, with or without EPS-1. Results demonstrated that EPS-1 attenuated the LPS-induced inflammatory response of the mucosa and the TER reduction, accompanying significantly alleviated morphological disruption of tight junction protein.

In this study, the expression of TNF-α, IFN-γ, and IL-6 in the colon tissues and Caco-2 monolayer were diminished dose-dependently by administrations of EPS-1, consistent with the reduction of DAI and histopathological scores. These results suggested that EPS-1 improved the imbalance of pro-inflammatory and anti-inflammatory cytokines in rat intestinal mucosa, and promoted self-repair of intestinal mucosa. In addition, pro-inflammatory cytokines, such as tumor necrosis factor TNF-α and IFN-γ cause dysregulation of barrier permeability during inflammation [3]. This underlying mechanism of the anti-inflammatory effect of EPS would be studied further.

Tight junction destruction plays a key role in the development and progression of inflammatory bowl disease (IBD) [39]. In previous animal experiment, it has been found that the congenital epithelial cell tight junction protein knockout mice appeared with intestinal pathological changes that were similar with that of IBD after birth, implying that TJ proteins were involved in the pathogenesis of IBD [40]. Various molecules, such as claudins, occludin, and E-canherin, are involved in maintaining this barrier [41,42]. However, loss of barrier integrity, initiated by bacteria or by treatment with a chemical, such as LPS, results in bacterial invasion and inflammation. A large number of studies have shown that restoring and maintaining intestinal mucosal barrier function were beneficial to improve the defensive function of intestinal mucosa, promote disease remission, and reduce relapse times of IBD [43,44]. Recently, Zhou X et al. found that EPS produced by *Lactobacillus plantarum* promoted epithelial barrier function and the expression of TJ protein and suggested that the regulation of the epithelial barrier function should be STAT3 dependent, which led to upregulation of Occludin and ZO-1 in the intestinal epithelial cells [45]. Our data also revealed that EPS-1 could improve the gut barrier function in vitro and in vivo and increased the expression of tight junction proteins. The specific mechanism of these actions need further study.

## 4. Materials and Methods 

### 4.1. Isolation and Purification of EPS

The *S. thermophilus* MN-BM-A01 was cultured in 12 g/100 mL skimmed milk at 37 °C for 24 h. After incubation, the EPS was isolated and purified as previously described [46]. Briefly, trichloroacetic acid was added to the culture to a final concentration of 4 g/100 mL, and the mixture was stirred for 30 min at room temperature. Cells and precipitated proteins were removed by centrifugation (8000× *g*, 4 °C, 10 min). Crude EPS was precipitated from the supernatant by addition of two volumes of cold ethanol overnight at 4 °C, and then collected by centrifugation at 8000× *g* for 10 min and dissolved in distilled water (50 g/100 mL). The solution was dialyzed using a dialysis bag (Mw cut-off 8 kDa to 14 kDa) against distilled water for 48 h at 4 °C with water replacement thrice a day. 

The crude EPS solution (0.2 g/mL, 10 mL) was fractionated with an anion exchange chromatography on a DEAE-FAST-FLOW column (50 × 500 mm), eluted with deionized water, 0.2 M, and 0.5 M NaCl solution at a flow rate of 1 mL/min. Every 5 mL of elution was collected automatically and the carbohydrate content was determined by phenol–sulfuric acid method. Peak fractions containing polysaccharides were pooled, dialyzed, and lyophilized. The purified EPS-1 solution (10 g/mL, 10 mL) was performed by an automatic polysaccharide gel purification system (Superdex 75, GE Healthcare, Boston, MA, USA) with column (1.6 × 80 cm), eluted with 0.5% NaCl solution at a flow rate of 0.2 mL/min. Every 2 mL of elution was collected automatically and the carbohydrate content was determined by the phenol–sulfuric acid method. Peak fractions containing polysaccharides were pooled, dialyzed, and lyophilized.

### 4.2. Molecular Mass Determination of EPS

The molecular mass of the purified EPS was measured by gel-permeation chromatography (GPC). The GPC system (Waters, Milford, MA, USA) consisted of three Waters Ultra-hydrogel 250, 1000 and 2000 (7.8 × 300 mm) columns in series. The column was eluted with 20 mM CH_3_COONH_4_ solution at a flow rate of 0.5 mL/min, and the injection volume of sample was 20 μL at an internal temperature of 40 °C.

### 4.3. Monosaccharide Analysis

The EPS (2 mg) was hydrolyzed with 1 mL trifluoroacetic acid (2 M) for 1.5 h. Excess trifluoroacetic acid was removed by rotary evaporation. The hydrolyzate was detected by GC-mass spectrometry (GC-MS) on a Hp-5 (Agilent Technology, Santa Clara, CA, USA) chromatographic column (300 × 0.25 × 0.32 mm) with hydrogen and air and helium at the flow rates of 30 and 400 mL/min, respectively, and helium was used as carrier gas at a flow rate of 1 mL/min. The standard sugars were rhamose, fucose, arabinose, xylose, mannose, glucose, and galactose.

### 4.4. Animals, Diets and Experimental Procedure.

Forty 6–8 weeks old pathogen-free male BALB/c mice were obtained from Vital River Laboratory Animal Technology Co. (Beijing, China). The animals were maintained on a 12-h-dark-light cycle and allowed free access to a basal diet and tap water under controlled temperatures (25 ± 2 °C). All experimental protocols (LA2016285) were approved by the Ethics Committee of Peking University Health Science Center and were conducted in accordance with the Guide for the Care and Use of Laboratory Animals.

After two weeks quarantine, the mice were divided into four groups and each group consisted of 10 animals. Untreated (control) groups were given 0.2 mL distilled water and received tap water orally for 14 days. For inducing chronic colitis, the rats in DSS treatment were administrated with 2.5 g/100 mL DSS (mol. wt 40 kD; TdB Consultancy, Uppsala, Sweden) in murine drinking water for seven days. A DSS alone group were given 0.2 mL distilled water from the day eight to 14. To study the therapeutic effect of EPS-1, this substance (20, 200 mg/kg body weight once per day) was orally administered from day eight to 14, respectively. Control mice were administered with PBS. Body weight loss was calculated as the percent difference between the original body weight and the actual body weight daily.

### 4.5. Evaluation of Colitis

In all animal daily weight, daily presence of gross blood, and daily stool consistency was determined. The DAI was determined by an investigator blinded to the protocol by scoring changes in weight, hemoccult positivity or gross bleeding, and stool consistency as the protocol previously described [47]. At the end of the experiment (day 14), after sacrificed, histological injury score of each colon was evaluated by morphological criteria as the protocol previously described [34]. The length of colons was measured and cut open longitudinally 1 cm of the distal colon for formalin fixing, paraffin embedding and HE staining. The paraffin sections of colon were graded by two blinded investigators with a range from zero to three as to the amount of inflammation (acute and chronic), depth of inflammation, and with a range from zero to four as to the amount of crypt damage or regeneration, as previously described [34]. The remaining colon tissue was used later to measure other tests.

### 4.6. Cell Lines and Monolayer Preparation

To investigate the effect of EPS on intestinal barrier dysfunction and pro-inflammatory cytokines, the Caco-2 cells were seeded on 24 well 12 mm polyester Transwell filters (Corning, Corning, NY, USA) with 0.4 μM pore size at a concentration of 2 × 10^5^ cells/Transwell. Caco-2 cells were cultured in Dulbecco’s Modified Eagle’s medium (DMEM) supplemented with 10% (*v*/*v*) foetal bovine serum (FBS) and 1 g/100 mL penicillin-streptomycin for 21 days. Caco-2 cells become a full monolayer with a mean TER exceeding 200 Ω·cm^2^. Keep on culturing until the TER exceeded 400 Ω·cm^2^ measured by a Millicell-Electrical Resistance System voltohmmeter (Millipore, Bedford, MA, USA) and the TER kept stable for three days until they formed a differentiated monolayer. For treatment, the Caco-2 cells were treated with LPS (10 μg/mL) without or with EPS-1 (2 mg/mL) for 24 h. EPS-1 (2 mg/mL) was added to the apical chamber of Transwell supports. Then, transepithelial electrical resistance (TER) values of all monolayers were measured with Millicell-ERS voltohmmeter. To measure dextran permeability, fluorescein isothiocyanate (FITC) dextran (4 kDa; 3 mg/mL) was added to the upper chamber without medium change. Aliquots were withdrawn from the lower chambers after 4 h and assayed for fluorescence at 515 nm with excitation at 492 nm. 

### 4.7. Measure of Cytokine Production in Colonic Mucosa and Cells

Colonic mucosa was cut into pieces and homogenised in ice-cold 100 mM Tris-HCl buffer, pH 7.0, containing a cocktail of protease inhibitors (Beyotime, Shanghai, China) and supplemented with 1 mM phenylmethanesulfonyl fluoride. After 12,744× *g* centrifugation at 4 °C for 10 min, the supernatant was collected and then quantified using bicinchoninic acid (BCA) protein assay reagent kit (Beyotime, Shanghai, China). The levels of TNF-α, IFN-γ, IL-4, IL-6, and IL-10 were measured using the enzyme-linked immunosorbent assay kits (R&D, Emeryville, CA, USA) according to the manufacturer’s recommendations. Briefly, anti-mouse cytokine antibodies were used to capture the proteins in supernatant, biotinylated polyclonal antibodies were used for detection. Color changes were determined at 450 nm. The amount of cytokines were determined based on the standard curve.

### 4.8. Western Blot Analysis

Colon tissues and Caco-2 cells were homogenated and lysed in lysis buffer supplemented with the protease and phosphatase inhibitor (PMSF, 0.5 mmol/L). Briefly, 100 mg tissue was homogenized in Radio Immunoprecipitation Assay (RIPA) lysis buffer (Beyotime) by using homogenizer. After homogenization, the supernatant was collected after a 5 min centrifugation at 17,346× *g*, 4 °C. 4 × 10^5^ Caco-2 cells were seeded in 6-Well Plate for 24 h. Then treated with LPS (10 μg/mL) without or with EPS-1 (2 mg/mL) incubation for another 24 h. The cells were then lysed and extracted protein with RIPA buffer by ultrasonication. Collected the supernatant by 12,744× *g* centrifugation, 4 °C, 10 min. The protein concentration was determined using BCA protein assay reagent kit (Beyotime), then the concentration of each sample was unified and the lysates were mixed with 4× sodium dodecyl sulfonate (SDS) sample loading buffer and denatured in boiling water for 5 min to denature the protein and then separated by 10~15 g/100 mL SDS-PAGE. Then the proteins were electro-transferred to PVDF member at 200 mA for 2 h. The membranes were incubated with primary antibodies: anti-β-actin (1:2500, Bioss, Beijing, China), Anti-Occludin (1:50,000, Abcam, Cambridge, MA, USA), anti-E-cadherin (1:1000, Abcam, USA), anti-Claudin1 (1:2000, Abcam, USA) for 12 h at 4 °C. After 5 times washed with 0.1 g/100 mL Tween-20 in PBS, then incubated with secondary antibodies (anti-rabbit and anti-mouse, 1:1000, Beyotime, Shanghai, China) for 1 h at room temperature. After being washed five times, the protein bands were visualized by ECL substrate (Millipore, Bedford, MA, USA) and Amersham Imager 600 (GE Healthcare, Boston, MA, USA). The density of the protein bands were quantified using Gel-pro software 4.0 (Media Cybernetics, Rockville, MD, USA) and β-actin was used as a control protein.

### 4.9. Statistical Analysis

All statistical analysis was done the statistical software Statistical Product and Service Solutions (version 19.0, SPSS Inc., Chicago, IL, USA) for Windows and results were expressed as mean ± standard deviation. The statistical significance of the differences among the experimental groups was evaluated using analysis of variance test and unpaired Student’s t-test. The level of statistical significance was defined as *p* < 0.05.

## 5. Conclusions

In this study, a water-soluble exopolysaccharide, designated as EPS-1, was isolated from MN-BM-A01 and purified. EPS-1 was a heteropolysaccharide, which was composed of rhamnose, glucose, galactose and mannose in an approximate molar ratio of 12.9:26.0:60.9:0.25. Our study confirmed the protective effects of purified EPS-1 on acute mouse colitis, mainly manifesting as the decrease of DAI and mitigated colonic epithelial cell injury. The protective effects might be related to the alleviation of intestinal inflammation and the improvement of mucosal barrier function. Thus, EPS-1 may be a preventive therapeutic agent for UC and a new health care product for intestinal health.

## Figures and Tables

**Figure 1 molecules-24-00513-f001:**
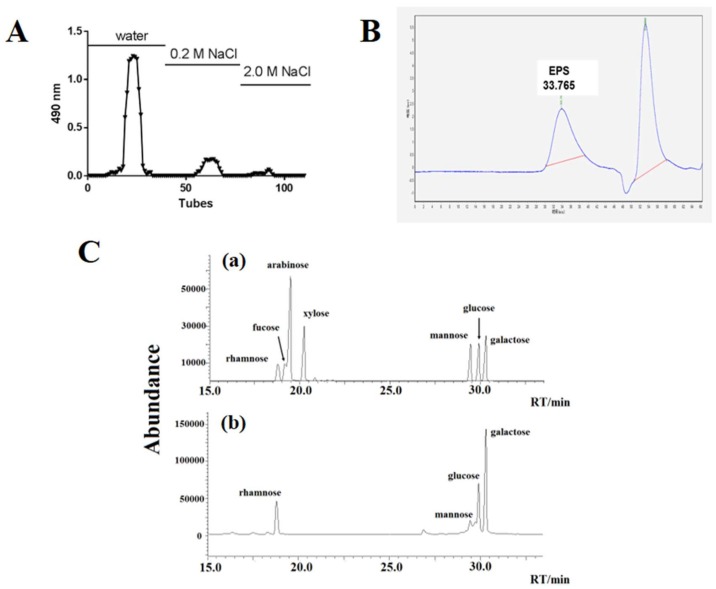
Isolation, molecular mass determination and monosaccharide composition of EPS from *S. thermophilus* MN-BM-A01. (**A**) Crude EPS separation profile by anion-exchange chromatography of DEAE Sepharose Fast flow. (**B**) GPC chromatogram of EPS-1. (**C**) Chromatogram of standard monosaccharides (**a**) and EPS-1 from *S. thermophilus* MN-BM-A01 monosaccharides (**b**) on chromatographic column.

**Figure 2 molecules-24-00513-f002:**
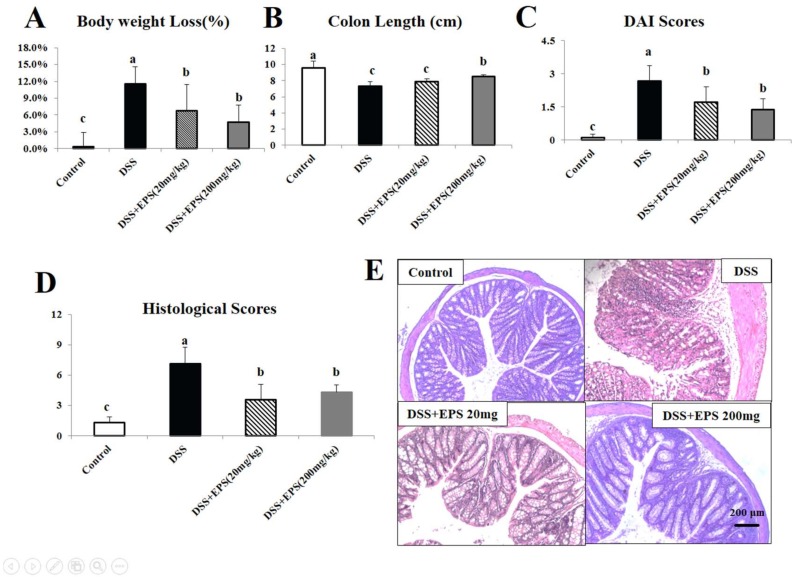
EPS-1 attenuates DSS-induced acute murine colitis. (**A**) Body weights loss, (**B**) variations of colon length, (**C**) disease activity index (DAI), and (**D**) histological scores of mice from each treatment group. (**E**) Representative HE staining colonic tissue from each treatment group, scale bars, 200 μm. Values with different superscript letters (a, b, c, d) are significantly different (*p* < 0.05).

**Figure 3 molecules-24-00513-f003:**
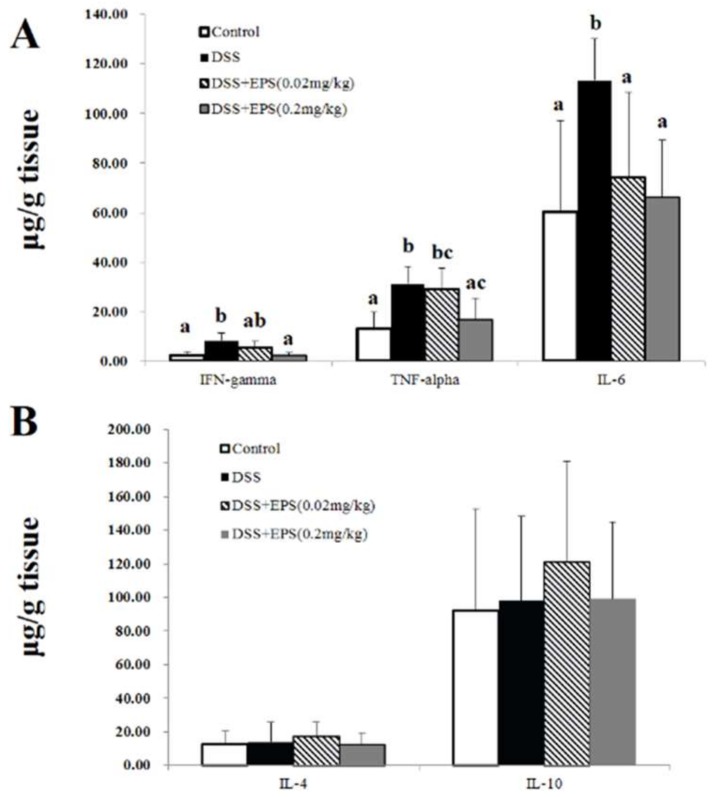
EPS-1 promoted a change in pro-inflammatory and anti-inflammatory cytokines. (**A**) The levels of pro-inflammatory cytokines in the colonic tissues from each treatment group. (**B**) The levels of anti-inflammatory cytokines in the colonic tissues from each treatment group. Values with different superscript letters (a, b, c) are significantly different (*p* < 0.05).

**Figure 4 molecules-24-00513-f004:**
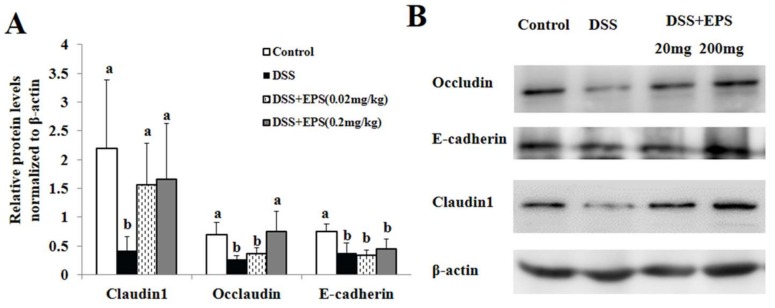
EPS-1 Relieved DSS-Induced Colonic Epithelial Tight Junction Disruption in Mice. (**A**) Relative protein levels of Occludin, E-cadherin and Claudin1 in the colon tissues normalized to β-actin. (**B**) Protein expression of Occludin, E-cadherin, and Claudin-1 in the colon tissues were analyzed by Western blot. Values with different superscript letters (a, b,) are significantly different (*p* < 0.05).

**Figure 5 molecules-24-00513-f005:**
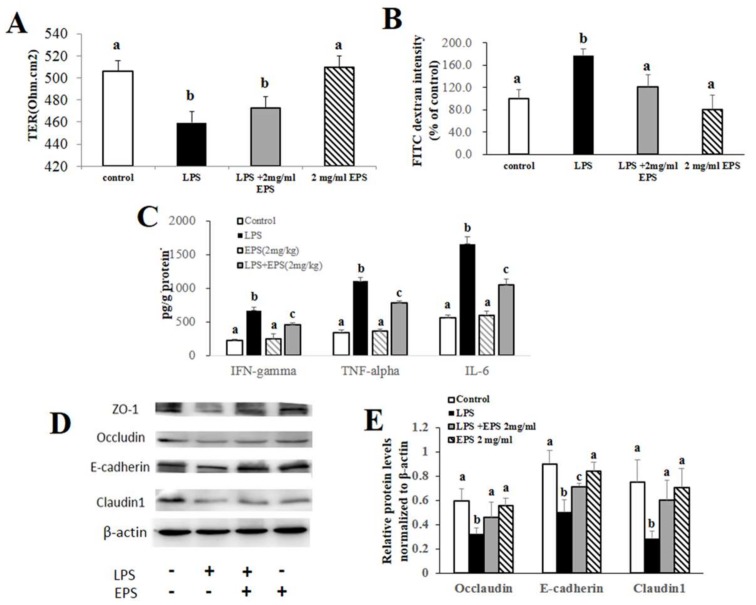
EPS-1 Protected intestinal barrier integrity from the disruption by LPS in Caco-2 monolayer. (**A**) The changes of transepithelial electrical resistance (TER) before and after treatment with LPS (10 μg/mL) and/or EPS-1 (2 mg/mL) for 24 h in Caco-2 monolayer. (**B**) FITC-dextran (4 kDa) permeability measurement in 21-day cultured Caco-2 monolayer after LPS and/or EPS-1 treatment. (**C**) The levels of pro-inflammatory cytokines in the Caco-2 cells from each treatment group. (**D**) Protein expression of Occludin, E-cadherin and Claudin1 in the Caco-2 cells from each treatment group. (**E**) Relative protein levels of Occludin, E-cadherin and Claudin1 in Caco-2 cells normalized to β-actin. Values with different superscript letters (a, b, c) are significantly different (*p* < 0.05).

**Table 1 molecules-24-00513-t001:** The monosaccharide composition of the EPS-1.

Sample NO.	Rhamnose (%)	Glucose (%)	Galactose (%)	Mannose (%)
**1**	12.5	25.8	61.4	0.25
**2**	13.4	26.1	60.2	0.23
**3**	12.7	26.0	61.1	0.27
Average proportion	12.9	26.0	60.9	0.25
Standard deviation	0.5	0.2	0.6	0.02

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
