# Peer review of "A Role of Exopolysaccharide Produced by *Streptococcus thermophilus* in the Intestinal Inflammation and Mucosal Barrier in Caco-2 Monolayer and Dextran Sulphate Sodium-Induced Experimental Murine Colitis"

_molecules, 2019, doi:10.3390/molecules24030513_

Round 1

Reviewer 1 Report

General comments:

1) Polysaccharides are rapidly metabolized by the organism. It is probably unlikely that they may have any effects as anti-inflammatory agents and on improving mucosal barrier function. To fully address the putative positive effects of polysaccharides, diet containing only monosaccharides such as rhamose, glucose and galactose shall have been performed.

2) It is not possible to rule out that other compounds than polysaccharides could have possible beneficial effects. In this respect, sample-to-sample variation was not adequately addressed. The mass spectra was not fully exploited to pint point possible other compounds.

3) The molecular mechanisms of polysaccharides and of monosaccharides to trigger anti-inflammatory response or improving mucosal barrier function need to be tackled to fully convince that such molecules could have any biological effects.

4) The coverage of literature was not critically assessed and some paragraphs lacked appropriate references.

5) Ethical concerns for animal uses were not addressed.

Minor comments:

6) Title: Change title since there is abbreviation DSS in the title.

7) Abstract: Delete abbreviation UC in the abstract since it was mentioned only once in the abstract.

8) Abstract: Delete abbreviation LPS and specify LPS

9) Abstract: Delete abbreviations TNF-α, IL-6 and IFN-γ and specify.

10) Abstract: Specify Caco-2

11) Introduction lines 37-38: The sentence is unclear:” which implies that lack of effective strategy was the determining factor for UC patients”.

12) Introduction, lines 39-41: Provide a reference to support the sentence “Although, precise mechanism of UC is still unclear, while there is no doubt 39 that intestinal mucosal barrier dysregulation and increased paracellular permeability play a critical role in pathogenesis of UC.”

13) Introduction, lines 41-42: Provide a reference to support “Tight junctions (TJ) of enterocyte regulates the integrity of intestinal barrier predominantly, which influence the paracellular and transcellular transport.”

14)Introduction lines 44-46: Provide several references to support “Many researches pointed out that, in UC patients, downregulations of tight junctions giving rise to the invasion of the colon wall with intestinal pathogens and toxins, which was considered as a vital event in the pathogenesis of intestinal inflammation.”

15) Introduction line 51: Specify abbreviation MLCK.

16) Introduction line 52: Specify abbreviation IL.

17) Introduction, lines 55-57: Provide references to support “Recently, increasingly evidences supported that probiotic supply beneficial effects for host health, especially in terms of intestinal function improvement and prevention of a variety of intestinal diseases including UC.”

18) Introduction, line 67: Specify MN-BM-A01 (CGMCC No. 11383).

19) Results, line 82: Provide mass determination by mass spectroscopy to confirm chromatography data and to ascertain distribution of molecular mass.

20 ) Results line 86: Chromatography profiles indicated the presence of fucose, and mannose in addition to rhamose, glucose and galactose that were not reported in the text.

21) Results line 87: There was two chromatograms in figure 1C that were not adequately described in figure legend.

22) Results line 86-87: Provide sample to sample errors on composition of monosaccharides.

23) Results line 107: Specify abbreviation HE.

24) Results line 112 and Fig 2E: It is unclear when histological scores were determined and so far there were no findings suggesting an increase of histological scores at day 14.

25) Results line 124: Change Figure 2A by Figure 3A.

26) Results line 129: Change Figure 2B by Figure 3B.

27) Results, lines 133 and 136: Change Figure 3 by Figure 4.

28) Results, line 133 and 136: Specify Figure 4A and Figure 4B.

29) Results, lines 150, 155, 156 and 162: Change Figure 4 by Figure 5.

30) Results, lines 150-165: There was no control on the viability of cells and it is unclear if the changes were related to toxicity of the treatment.

31) Results, line 156, Figure 5D: Provide quantitative estimation of Western Blot profiles.

32) Results, line 162, Figure 5C: Indicate on the ordinate what correspond the numbers.

33) Discussion, lines 175-176: Provide one reference to support that “Ulcerative colitis (UC) is a set of complicated chronic inflammatory and ulceration conditions of the colonic mucosa, accompanied by clinical symptoms.”

33) Discussion, lines 176-177: Provide references to support that “Recently, increasingly evidences supported that probiotics conferred a lot of health benefits, including UC prevention”

34) Discussion, lines 176-177: Provide references to support that “EPS may play a role in this prevention process. However, there is no literature reporting on the intervention effects of purified EPS in experimental colitis. ”

35) Discussion, lines 185-186: Provide references to support that S. thermophilus is the most important dairy starter. The exopolysaccharides of S. thermophilus can improve the properties of the dairy product and confer benefcial health effects.

36) Discussion, lines 190-192: At this stage no firm conclusion can be drawn on the composition of exopolysaccharide due to lack of sample to sample determinations and beside two monosacharrides fucose and mannose were omitted (Figure 1C) in the composition. The sentence “In this study, the EPS-1 produced by S. thermophilus MN-BM-A01 was a representative heteropolysaccharide, consisting of rhamnose, glucose and galactose in a molar ratio of 2.6: 1.3: 6.1.” shall be deleted.

37) Discussion, line 193: The sentence” UC is a chronic inflammatory disease of large intestine.” Shall be deleted, it is stated in lines 175-176.

38) Discussion, lines 193,194: Provide one reference to support that “The occurrence and development of UC were strongly associated with the imbalance of pro-inflammatory and anti-inflammatory cytokines”.

39) Discussion, lines 195,196: Provide references to support that “Previous reports showed that a UC-like mouse model of DSS-induced colitis had neutrophil accumulation and increased expression of pro-inflammatory in the colon.”

40) Discussion, lines 195,194: There were no experimental evidence that cytokines were released by macrophages. The sentence “Our study also demonstrated that production of macrophage-derived cytokines TNF-α, IFN-γ and IL-6” shall be modified or deleted.

41) Discussion, lines 204-206: So far the experiments were performed on mice and not on rats: The sentence” Our study also demonstrated that production of macrophage-derived cytokines TNF-α, IFN-γ and IL-6 increased significantly, but anti-inflammatory cytokines (IL-4 and IL-10)” shall be deleted or modified.

Discussion, line 208-213: Specify probiotic in several sentences “Although therapeutic effects of probiotics on colitis were strain-specific and dose-dependant, the anti-inflammatory effects of probiotics play a decisive role [27, 28].P revious study has indicated that the expression of TNF-α, IL-1β, and IL-6 in the colon tissues was diminished dose-dependently by a mixture of three potential probiotic strains in DSS induced colitis[29]. Pan T. et.al also illustrated that an appropriate dose of probiotics strain can prevent intestinal damage in mice with DSS-induced colitis by inhibiting the expression of inflammatory cytokines [30].”

42) Discussion, lines 213-218: The sentences “In our study, EPS-1 treatment attenuated the release of TNF-α, IFN-γ and IL-6 significantly, suggesting that this effect is the primary cause of anti-colitic activity of EPS. The inhibition effect or pro-inflammatory cytokines was further confirmed in Caco-2 cell lines treated by LPS with or without EPS-1. Results demonstrated that EPS-1 attenuated the LPS-induced inflammatory response of the mucosa and the TER reduction,accompanying significantly alleviated morphological disruption of tight junction protein.” Shall be softened since it is unlikely that polysaccharides could have an anti-inflammatory effect, since they are rapidly metabolized.

43) Discussion lines 237-238: The sentence “Our data also revealed that EPS-1 facilitated the gut barrier function in vitro and in vivo and increased the expression of tight junction proteins, which might be caused by the suppressing of pro-inflammatory cytokines” shall be softened.

44) Conclusion, lines 242-248: The whole paragraph “In this study, a water-soluble exopolysaccharide, designated as EPS-1, was isolated from MN243

BM-A01 and purified. EPS-1 was a heteropolysaccharide, which was composed of rhamnose, glucose and galactose in a molar ratio of 2.6: 1.3: 6.1. Our study confirmed the protective effects of purified EPS-1 on acute mouse colitis, mainly manifesting as the decrease of DAI and mitigated colonic epithelial cell injury. The protective effects might be related to the alleviation of intestinal inflammation and the improvement of mucosal barrier function. Thus, EPS-1 may be a preventive therapeutic agent for UC and a new health care product for intestinal health.” Is already stated in the discussion and shall be deleted.

45) Materials and Methods, lines  251, 253, 256, 285, 305, 306 and 326: Specify units in %.

46) Materials and Methods, line 254: Indicate in which buffer (composition and final pH) was added trifluoroacetic acid.

47) Materials and methods, line 263: Provide more information on further purification.

48) Materials and Methods, line 269 replace CH3COONH4 by CH3COONH4

49) Materials and Methods, line 272: Specify TFA.

50) Materials and Methods, lines 279: Provide information on ethical committee.

51) Materials and Methods, line 279: Specify BALB/c.

52) Materials and Methods, line 295: Specify when the length of colons was measured.

53) Materials and Methods, line 315: Provide more information on  preparation of colonic mucosa.

54) Materials and Methods, line 317: Provide more information on determination of TNF-α, IFN-γ, IL-4, IL-6 and IL-10.

55) Materials and Methods, line 320: Provide more information on  preparation of colon tissues.

56) Materials and Methods, line 320: Provide more information on  preparation of Caco-2 cells, how long ere they incubated, etc…

57) Materials and Methods, line 322: Provide more information how proteins were extracted.

58) References contained mostly relatively old papers.

Author Response

Response to Reviewer 1 Comments

General comments:

1) Polysaccharides are rapidly metabolized by the organism. It is probably unlikely that they may have any effects as anti-inflammatory agents and on improving mucosal barrier function. To fully address the putative positive effects of polysaccharides, diet containing only monosaccharides such as rhamose, glucose and galactose shall have been performed.

Response 1:

Thank you for your suggestions. Exopolysaccharides (EPS) are polymers synthesised by a range of bacterial groups. In general, EPS was not easy to digest which was shown to protect the bacterial cell from environmental stresses, such as human gastric and pancreatic enzymes, bile salts and varying pH [1]. A previous report has improved that EPS produced by Lactococcus lactis ssp. cremoris B40, Lactobacillus sakei 0–1, Streptococcus thermophilus SFi20, and Lactobacillus helveticus Lh59 all retain integrity during gastric transit [2].

For this reason, the undigested EPS can reach the lower intestine and positively impact on the gut microbiome and indirectly modulate the immune system. It was well documented that dendritic cells (DC) sample luminal microbes and microbial components as antigens, such as EPS, resulting in augmentation of natural killer cell activity and modulate inflammatory cytokine expression including interleukins, interferons and tumour necrosis factor [3, 4]. In this study, anti-inflammatory effect of EPS may be related to this mechanism and we will study further next.

Reference:

1.         Ryan, P. M.; Ross, R. P.; Fitzgerald, G. F.; Caplice, N. M.; Stanton, C., Sugar-coated: exopolysaccharide producing lactic acid bacteria for food and human health applications. Food Funct 2015, 6, (3), 679-693.

2.         Ruijssenaars, H. J.; Stingele, F.; Hartmans, S., Biodegradability of food-associated extracellular polysaccharides. Curr Microbiol 2000, 40, (3), 194-199.

3.         Farache, J.; Koren, I.; Milo, I.; Gurevich, I.; Kim, K. W.; Zigmond, E.; Furtado, G. C.; Lira, S. A.; Shakhar, G., Luminal Bacteria Recruit CD103(+) Dendritic Cells into the Intestinal Epithelium to Sample Bacterial Antigens for Presentation. Immunity 2013, 38, (3), 581-595.

4.         Arena, A.; Maugeri, T. L.; Pavone, B.; Iannello, D.; Gugliandolo, C.; Bisignano, G., Antiviral and immunoregulatory effect of a novel exopolysaccharide from a marine thermotolerant Bacillus licheniformis. International immunopharmacology 2006, 6, (1), 8-13.

2) It is not possible to rule out that other compounds than polysaccharides could have possible beneficial effects. In this respect, sample-to-sample variation was not adequately addressed. The mass spectra was not fully exploited to pint point possible other compounds.

Response 2:

We are sorry for the unclear description. Just as your concern, the purity of EPS is very important for our study. The EPS was isolated and purified as previously described [5]. There were four steps in our study to ensure the purity of EPS. Firstly, after fermentation, trichloroacetic acid (TCA) was added to the culture to a final concentration of 4% (w/v), and the mixture was stirred for 30 min at room temperature. Cells and precipitated proteins were removed by centrifugation (8,000 × g, 4 ◦C, 10 min). Secondly, crude EPS was precipitated from the supernatant by addition of 2 volumes of cold ethanol stored at 4◦C for 24 h. Crude EPS was collected by centrifugation at 8,000 × g for 10 min. Most of the rest proteins, liposoluble and other components were removed. Thirdly, crude EPS solution was dialyzed using a dialysis bag (Mw cut-off 8 kDa to 14 kDa) against distilled water for 48 h at 4 °C to remove the low molecular weight components. Fourthly, before the crude EPS solution was fractionated with an anion exchange chromatography, we tested the sample again for protein, amino acid and polypeptide content to rule out interference from other compounds. We have revised the original manuscript in this part to make it easier for readers to understand.

The average molecular weights of the purified EPS were measured by gel permeation chromatography (GPC) rather than mass spectroscopy. GPC is a commonly used method for determining the average molecular weight of polysaccharides [5, 6]. Standard dextrans (4, 10, 32, 100, 500kDa, Fluka Chemical Co., Buchs, Switzerland) were passed through Waters Ultra-hydrogel 250, 1000 and 2000 (7.8 × 300 mm) columns in series. The column was eluted with 20 mM CH3COONH4 solution at a flow rate of 0.5 mL/min, and the injection volume of sample was 20 μL at an internal temperature of 40 °C. The standard curve equation Log Mw = -0.1741x + 11.505 (R2 = 0.9913), where Mw is the peak molecular weight and x is the retention time (figure 1 below).

Figure 1 Standard curve of dextrans measured by gel permeation chromatography

Thank you for your suggestions. In the revised manuscript, molar ratio of rhamnose, glucose, galactose and mannose was changed to 12.9: 26.0: 60.8: 0.25. We analyzed the three extracted EPS samples. The approximate molar ratio of the sugar monomers was shown in revised figure 1C and the table below. The relative proportion of four different sugar monomers varied slightly between samples. The proportion of mannose was very low (0.25%).

Sample NO.

Rhamnose(%)

Glucose(%)

Galactose(%)

Mannose(%)

1

12.54

25.81

61.40

0.25

2

13.43

26.14

60.20

0.23

3

12.68

25.97

61.08

0.27

Average proportion

12.88

25.97

60.8%

0.25

Standard deviation

0.48

0.16

0.62

0.02

Reference:

5.         Zhang, J.; Cao, Y.; Wang, J.; Guo, X.; Zheng, Y.; Zhao, W.; Mei, X.; Guo, T.; Yang, Z., Physicochemical characteristics and bioactivities of the exopolysaccharide and its sulphated polymer from Streptococcus thermophilus GST-6. Carbohydrate polymers 2016, 146, 368-75.

3) The molecular mechanisms of polysaccharides and of monosaccharides to trigger anti-inflammatory response or improving mucosal barrier function need to be tackled to fully convince that such molecules could have any biological effects.

Response 3:

Thank you for your suggestions. As discussed above, EPS was not easy to digest, the undigested intact EPS can reach the intestine and positively impact on the gut microbiome and indirectly modulate the immune system. Dendritic cells (DC) sampled EPS as antigens, resulting in immunoregulation of inflammatory cytokine expression including interleukins, interferons and tumour necrosis factor [3, 4]. In this study, anti-inflammatory effect of EPS maybe related to this mechanism and we will study further next.

Reference:

3.         Farache, J.; Koren, I.; Milo, I.; Gurevich, I.; Kim, K. W.; Zigmond, E.; Furtado, G. C.; Lira, S. A.; Shakhar, G., Luminal Bacteria Recruit CD103(+) Dendritic Cells into the Intestinal Epithelium to Sample Bacterial Antigens for Presentation. Immunity 2013, 38, (3), 581-595.

4.         Arena, A.; Maugeri, T. L.; Pavone, B.; Iannello, D.; Gugliandolo, C.; Bisignano, G., Antiviral and immunoregulatory effect of a novel exopolysaccharide from a marine thermotolerant Bacillus licheniformis. International immunopharmacology 2006, 6, (1), 8-13.

4) The coverage of literature was not critically assessed and some paragraphs lacked appropriate references.

Response 4:

We are sorry for the careless and these mistakes. We have corrected these mistakes in revised manuscript.

5) Ethical concerns for animal uses were not addressed.

Response 5:

 Thank you for your suggestions. Ethical concerns for animal was shown below.

Minor comments:

6) Title: Change title since there is abbreviation DSS in the title.

Response We are very sorry for the mistakes and we have corrected the text accordingly.

7) Abstract: Delete abbreviation UC in the abstract since it was mentioned only once in the abstract.

Response We are very sorry for the mistakes and we have corrected the text accordingly.

8) Abstract: Delete abbreviation LPS and specify LPS

Response: We are very sorry for the mistakes and we have corrected the text accordingly.

9) Abstract: Delete abbreviations TNF-α, IL-6 and IFN-γ and specify.

Response: We are very sorry for the mistakes and we have corrected the text accordingly.

10) Abstract: Specify Caco-2

Response: Thank you for your suggestions. But as far as I know, the Caco-2 cell line is a continuous line of human epithelial colorectal adenocarcinoma cells. “Caco-2” is not an abbreviated form.

11) Introduction lines 37-38: The sentence is unclear:” which implies that lack of effective strategy was the determining factor for UC patients”.

Response: We are sorry and we have modified sentence to make it clear.

12) Introduction, lines 39-41: Provide a reference to support the sentence “Although, precise mechanism of UC is still unclear, while there is no doubt 39 that intestinal mucosal barrier dysregulation and increased paracellular permeability play a critical role in pathogenesis of UC.”

Response: Thank you for your advice. We have added a new reference to support the sentence.

13) Introduction, lines 41-42: Provide a reference to support “Tight junctions (TJ) of enterocyte regulates the integrity of intestinal barrier predominantly, which influence the paracellular and transcellular transport.”

Response: Thank you for your advice. We have added a new reference to support the sentence.

14)Introduction lines 44-46: Provide several references to support “Many researches pointed out that, in UC patients, downregulations of tight junctions giving rise to the invasion of the colon wall with intestinal pathogens and toxins, which was considered as a vital event in the pathogenesis of intestinal inflammation.”

Response: Thank you for your advice. We have added references to support the sentence.

15) Introduction line 51: Specify abbreviation MLCK.

Response: We are very sorry for the mistakes and we have corrected the text accordingly.

16) Introduction line 52: Specify abbreviation IL.

Response: We are very sorry for the mistakes and we have corrected the text accordingly.

17) Introduction, lines 55-57: Provide references to support “Recently, increasingly evidences supported that probiotic supply beneficial effects for host health, especially in terms of intestinal function improvement and prevention of a variety of intestinal diseases including UC.”

Response: Thank you for your advice. We have added references to support the sentence.

18) Introduction, line 67: Specify MN-BM-A01 (CGMCC No. 11383).

Response: Thank you for your suggestions. However, MN-BM-A01 is the full name of the strain rather than the abbreviation. “CGMCC No. 11383” is the strain preservation number in China general microbiological culture collection center.

19) Results, line 82: Provide mass determination by mass spectroscopy to confirm chromatography data and to ascertain distribution of molecular mass.

Response: The average molecular weights of the purified EPS were measured by gel permeation chromatography (GPC) rather than mass spectroscopy. GPC is a commonly used method for determining the average molecular weight of polysaccharides [5, 6]. Standard dextrans (4, 10, 32, 100, 500kDa, Fluka Chemical Co., Buchs, Switzerland) were passed through Waters Ultra-hydrogel 250, 1000 and 2000 (7.8 × 300 mm) columns in series. The column was eluted with 20 mM CH3COONH4 solution at a flow rate of 0.5 mL/min, and the injection volume of sample was 20 μL at an internal temperature of 40 °C. The standard curve equation Log Mw = -0.1741x + 11.505 (R2 = 0.9913), where Mw is the peak molecular weight and x is the retention time (figure 1 below).

Figure 1 standard curve of dextrans measured by gel permeation chromatography

Reference:

5.         Zhang, J.; Cao, Y.; Wang, J.; Guo, X.; Zheng, Y.; Zhao, W.; Mei, X.; Guo, T.; Yang, Z., Physicochemical characteristics and bioactivities of the exopolysaccharide and its sulphated polymer from Streptococcus thermophilus GST-6. Carbohydrate polymers 2016, 146, 368-75.

6.         Striegel, A. M.; Timpa, J. D., Molecular Characterization of Polysaccharides Dissolved in Me(2)Nac-Licl by Gel-Permeation Chromatography. Carbohyd Res 1995, 267, (2), 271-290.

20 ) Results line 86: Chromatography profiles indicated the presence of fucose, and mannose in addition to rhamose, glucose and galactose that were not reported in the text.

Response: We are sorry for the unclear description and thanks for your suggestions. We added the proportion of mannose and provided a new chromatography profile (fig 1C) in revised manuscript. EPS-1 was composed of four different sugar monomers including rhamnose, glucose, galactose and mannose. We analyzed the three extracted EPS samples. The approximate molar ratio of the sugar monomers was shown in table below. The relative proportion of four different sugar monomers varied slightly between samples. The proportion of mannose was very low (0.25%).

Sample NO.

Rhamnose(%)

Glucose(%)

Galactose(%)

Mannose(%)

1

12.54

25.81

61.40

0.25

2

13.43

26.14

60.20

0.23

3

12.68

25.97

61.08

0.27

Average proportion

12.88

25.97

60.8%

0.25

Standard deviation

0.48

0.16

0.62

0.02

21) Results line 87: There was two chromatograms in figure 1C that were not adequately described in figure legend.

Response: We are sorry for the unclear description. We provided chromatography profile (fig 1C) and modified the figure legend in revised manuscript.

22) Results line 86-87: Provide sample to sample errors on composition of monosaccharides.

Response: We analyzed the three extracted EPS samples. The approximate molar ratio of the sugar monomers was shown in table below.

Sample NO.

Rhamnose(%)

Glucose(%)

Galactose(%)

Mannose(%)

1

12.54

25.81

61.40

0.25

2

13.43

26.14

60.20

0.23

3

12.68

25.97

61.08

0.27

Average proportion

12.88

25.97

60.8%

0.25

Standard deviation

0.48

0.16

0.62

0.02

23) Results line 107: Specify abbreviation HE.

Response: We are very sorry for the mistakes and we have corrected the text accordingly.

24) Results line 112 and Fig 2E: It is unclear when histological scores were determined and so far there were no findings suggesting an increase of histological scores at day 14.

Response: We are sorry for the unclear description. At the end of the experiment (day 14), after sacrificed, histological injury score of each colon was evaluated by morphological criteria as the protocol previously described. The length of colons was measured and cut open longitudinally 1 cm of the distal colon for formalin fixing, paraffin embedding and HE staining. The paraffin sections of colon were graded by two blinded investigators with a range from 0 to 3 as to amount of inflammation (acute and chronic), depth of inflammation and with a range from 0 to 4 as to the amount of crypt damage or regeneration as previously described [7]. Histological scores was the sum of these scores.

Reference:

7.    Dieleman, L. A.; Palmen, M. J.; Akol, H.; Bloemena, E.; Pena, A. S.; Meuwissen, S. G.; Van Rees, E. P., Chronic experimental colitis induced by dextran sulphate sodium (DSS) is characterized by Th1 and Th2 cytokines. Clinical and experimental immunology 1998, 114, (3), 385-91.

25) Results line 124: Change Figure 2A by Figure 3A

Response: We are very sorry for the mistakes and we have corrected the text accordingly.

26) Results line 129: Change Figure 2B by Figure 3B.

Response: We are very sorry for the mistakes and we have corrected the text accordingly.

27) Results, lines 133 and 136: Change Figure 3 by Figure 4.

Response: We are very sorry for the mistakes and we have corrected the text accordingly.

28) Results, line 133 and 136: Specify Figure 4A and Figure 4B.

Response: We are very sorry for the mistakes and we have corrected the text accordingly.

29) Results, lines 150, 155, 156 and 162: Change Figure 4 by Figure 5.

Response: We are very sorry for the mistakes and we have corrected the text accordingly.

30) Results, lines 150-165: There was no control on the viability of cells and it is unclear if the changes were related to toxicity of the treatment.

Response: We are sorry for the unclear description. In cellular experiments, EPS (2mg/mL) treatment alone was regarded as control to measure the toxicity of EPS.

31) Results, line 156, Figure 5D: Provide quantitative estimation of Western Blot profiles.

Response: Thank you for your advice. We have added a new fig.5E to show the quantitative analysis of Western Blot profiles

32) Results, line 162, Figure 5C: Indicate on the ordinate what correspond the numbers.

Response: We are very sorry for the mistakes and we have added the ordinate in fig. 5C.

33) Discussion, lines 175-176: Provide one reference to support that “Ulcerative colitis (UC) is a set of complicated chronic inflammatory and ulceration conditions of the colonic mucosa, accompanied by clinical symptoms.”

Response: Thank you for your advice. We have added a new reference to support the sentence.

33) Discussion, lines 176-177: Provide references to support that “Recently, increasingly evidences supported that probiotics conferred a lot of health benefits, including UC prevention”

Response: Thank you for your advice. We have added new references to support the sentence.

34) Discussion, lines 176-177: Provide references to support that “EPS may play a role in this prevention process. However, there is no literature reporting on the intervention effects of purified EPS in experimental colitis. ”

Response: Thank you for your advice. We have added new references to support the sentence.

35) Discussion, lines 185-186: Provide references to support that S. thermophilus is the most important dairy starter. The exopolysaccharides of S. thermophilus can improve the properties of the dairy product and confer benefcial health effects.

Response: Thank you for your advice. We have added new references to support the sentence.

36) Discussion, lines 190-192: At this stage no firm conclusion can be drawn on the composition of exopolysaccharide due to lack of sample to sample determinations and beside two monosacharrides fucose and mannose were omitted (Figure 1C) in the composition. The sentence “In this study, the EPS-1 produced by S. thermophilus MN-BM-A01 was a representative heteropolysaccharide, consisting of rhamnose, glucose and galactose in a molar ratio of 2.6: 1.3: 6.1.” shall be deleted.

Response: We are sorry for the unclear description and thanks for your suggestions. We added the proportion of mannose and provided a new chromatography profile (fig 1C) in revised manuscript. We analyzed the three extracted EPS samples. The approximate molar ratio of the sugar monomers was shown in table below. In summary, EPS was composed of different sugar monomers including rhamnose, glucose, galactose and mannose in an approximate molar ratio of 12.9 : 26.0 : 60.8 : 0.25

Sample NO.

Rhamnose(%)

Glucose(%)

Galactose(%)

Mannose(%)

1

12.54

25.81

61.40

0.25

2

13.43

26.14

60.20

0.23

3

12.68

25.97

61.08

0.27

Average proportion

12.88

25.97

60.8%

0.25

Standard deviation

0.48

0.16

0.62

0.02

37) Discussion, line 193: The sentence” UC is a chronic inflammatory disease of large intestine.” Shall be deleted, it is stated in lines 175-176.

Response: We are very sorry for the mistakes and we have corrected the text accordingly.

38) Discussion, lines 193,194: Provide one reference to support that “The occurrence and development of UC were strongly associated with the imbalance of pro-inflammatory and anti-inflammatory cytokines”.

Response: Thank you for your advice. We have added a new reference to support the sentence.

39) Discussion, lines 195,196: Provide references to support that “Previous reports showed that a UC-like mouse model of DSS-induced colitis had neutrophil accumulation and increased expression of pro-inflammatory in the colon.”

Response: Thank you for your advice. We have added new references to support the sentence.

40) Discussion, lines 195,194: There were no experimental evidence that cytokines were released by macrophages. The sentence “Our study also demonstrated that production of macrophage-derived cytokines TNF-α, IFN-γ and IL-6” shall be modified or deleted.

Response: We are very sorry for the mistakes and we have corrected the text accordingly.

41) Discussion, lines 204-206: So far the experiments were performed on mice and not on rats: The sentence” Our study also demonstrated that production of macrophage-derived cytokines TNF-α, IFN-γ and IL-6 increased significantly, but anti-inflammatory cytokines (IL-4 and IL-10)” shall be deleted or modified.

Discussion, line 208-213: Specify probiotic in several sentences “Although therapeutic effects of probiotics on colitis were strain-specific and dose-dependant, the anti-inflammatory effects of probiotics play a decisive role [27, 28].P revious study has indicated that the expression of TNF-α, IL-1β, and IL-6 in the colon tissues was diminished dose-dependently by a mixture of three potential probiotic strains in DSS induced colitis[29]. Pan T. et.al also illustrated that an appropriate dose of probiotics strain can prevent intestinal damage in mice with DSS-induced colitis by inhibiting the expression of inflammatory cytokines [30].”

Response: We are very sorry for the mistakes and we have corrected the text accordingly.

42) Discussion, lines 213-218: The sentences “In our study, EPS-1 treatment attenuated the release of TNF-α, IFN-γ and IL-6 significantly, suggesting that this effect is the primary cause of anti-colitic activity of EPS. The inhibition effect or pro-inflammatory cytokines was further confirmed in Caco-2 cell lines treated by LPS with or without EPS-1. Results demonstrated that EPS-1 attenuated the LPS-induced inflammatory response of the mucosa and the TER reduction,accompanying significantly alleviated morphological disruption of tight junction protein.” Shall be softened since it is unlikely that polysaccharides could have an anti-inflammatory effect, since they are rapidly metabolized.

Response:

Thank you for your suggestions. Exopolysaccharides (EPS) are polymers synthesised by a range of bacterial groups. In general, EPS was not easy to digest which was shown to protect the bacterial cell from environmental stresses, such as human gastric and pancreatic enzymes, bile salts and varying pH [1]. A previous report has improved that EPS produced by Lactococcus lactis ssp. cremoris B40, Lactobacillus sakei 0–1, Streptococcus thermophilus SFi20, and Lactobacillus helveticus Lh59 all retain integrity during gastric transit [2].

For this reason, the undigested EPS can reach the lower intestine and positively impact on the gut microbiome and indirectly modulate the immune system. It was well documented that dendritic cells (DC) sample luminal microbes and microbial components as antigens, such as EPS, resulting in augmentation of natural killer cell activity and modulate inflammatory cytokine expression including interleukins, interferons and tumour necrosis factor [3, 4]. In this study, anti-inflammatory effect of EPS may be related to this mechanism and we will study further next.

Reference:

1.         Ryan, P. M.; Ross, R. P.; Fitzgerald, G. F.; Caplice, N. M.; Stanton, C., Sugar-coated: exopolysaccharide producing lactic acid bacteria for food and human health applications. Food Funct 2015, 6, (3), 679-693.

2.         Ruijssenaars, H. J.; Stingele, F.; Hartmans, S., Biodegradability of food-associated extracellular polysaccharides. Curr Microbiol 2000, 40, (3), 194-199.

3.         Farache, J.; Koren, I.; Milo, I.; Gurevich, I.; Kim, K. W.; Zigmond, E.; Furtado, G. C.; Lira, S. A.; Shakhar, G., Luminal Bacteria Recruit CD103(+) Dendritic Cells into the Intestinal Epithelium to Sample Bacterial Antigens for Presentation. Immunity 2013, 38, (3), 581-595.

4.         Arena, A.; Maugeri, T. L.; Pavone, B.; Iannello, D.; Gugliandolo, C.; Bisignano, G., Antiviral and immunoregulatory effect of a novel exopolysaccharide from a marine thermotolerant Bacillus licheniformis. International immunopharmacology 2006, 6, (1), 8-13.

43) Discussion lines 237-238: The sentence “Our data also revealed that EPS-1 facilitated the gut barrier function in vitro and in vivo and increased the expression of tight junction proteins, which might be caused by the suppressing of pro-inflammatory cytokines” shall be softened.

Response: Thank you for your suggestions. We have made corresponding modifications according to your opinions.

44) Conclusion, lines 242-248: The whole paragraph “In this study, a water-soluble exopolysaccharide, designated as EPS-1, was isolated from MN-BM-A01 and purified. EPS-1 was a heteropolysaccharide, which was composed of rhamnose, glucose and galactose in a molar ratio of 2.6: 1.3: 6.1. Our study confirmed the protective effects of purified EPS-1 on acute mouse colitis, mainly manifesting as the decrease of DAI and mitigated colonic epithelial cell injury. The protective effects might be related to the alleviation of intestinal inflammation and the improvement of mucosal barrier function. Thus, EPS-1 may be a preventive therapeutic agent for UC and a new health care product for intestinal health.” Is already stated in the discussion and shall be deleted.

Response: Thank you for your suggestions. We have made corresponding modifications according to your opinions.

45) Materials and Methods, lines  251, 253, 256, 285, 305, 306 and 326: Specify units in %.

Response: We are very sorry for the mistakes and we have corrected the text accordingly.

46) Materials and Methods, line 254: Indicate in which buffer (composition and final pH) was added trifluoroacetic acid.

Response: We are sorry for the unclear description. Trichloroacetic acid was added to the culture to a final concentration of 4% (w/v), and the mixture was stirred for 30 min at room temperature. Cells and precipitated proteins were removed by centrifugation. There was no buffer added to the culture.

47) Materials and methods, line 263: Provide more information on further purification.

Response: We are sorry for the unclear description, and we have modified section to make it clearly. The purified EPS-1 solution (10 g/mL, 10 mL) was performed by an automatic polysaccharide gel purification system (Superdex75) with column (1.6 × 80 cm), eluted with 0.5 % NaCl solution at a flow rate of 0.2 mL/min. Every 2 mL of elution was collected automatically and the carbohydrate content was determined by phenol–sulfuric acid method. Peak fractions containing polysaccharides were pooled, dialyzed, and lyophilized.

48) Materials and Methods, line 269 replace CH3COONH4 by CH3COONH4

Response: We are very sorry for the mistakes and we have corrected the text accordingly.

49) Materials and Methods, line 272: Specify TFA.

Response: We are very sorry for the mistakes and we have corrected the text accordingly.

50) Materials and Methods, lines 279: Provide information on ethical committee.

Response: Thank you for your suggestions. Ethical concerns for animal was shown below.

51) Materials and Methods, line 279: Specify BALB/c.

Response: Thank you for your suggestions. But as far as I know, “BALB/c” is not an abbreviated form.In 1974, this substrain of mouse were returned to The Jackson Laboratory and were named BALB/c. BALB/c mice are distributed globally, and are among the most widely used inbred strains used in animal experimentation.

Reference:

A Brief History of the Two Substrains of BALB/c, BALB/cJ, and BALB/cByJ". Jax Mice Literature. Jackson Laboratory. Archived from the original on 19 June 2011. 2010-09-30.

52) Materials and Methods, line 295: Specify when the length of colons was measured.

Response: Thank you for your advice. At the end of the experiment (day 14), after sacrificed , the length of colons was measured, and we have corrected the text accordingly.

53) Materials and Methods, line 315: Provide more information on  preparation of colonic mucosa.

Response: Thank you for your advice. We have added information on preparation of colonic mucosa accordingly.

54) Materials and Methods, line 317: Provide more information on determination of TNF-α, IFN-γ, IL-4, IL-6 and IL-10.

Response: Thank you for your advice. We have added information on determination of cytokines accordingly.

55) Materials and Methods, line 320: Provide more information on  preparation of colon tissues.

Response: Thank you for your advice. We have added information on preparation of colon tissues accordingly.

56) Materials and Methods, line 320: Provide more information on  preparation of Caco-2 cells, how long ere they incubated, etc…

Response: Thank you for your advice. We have added information on preparation of colon tissues accordingly.

57) Materials and Methods, line 322: Provide more information how proteins were extracted.

Response: Thank you for your advice. We have added information on preparation of colon tissues accordingly.

58) References contained mostly relatively old papers.

Response: Thank you for your advice. We have added some relative newer references in the revised manuscript. 

Reviewer 2 Report

The manuscript (Ref. no.molecules-425482) have described the effects of exopolysaccharide isolated from Streptococcus thermophilus on the DSS-induced colitis and tight junction disruption  in mice as well as pro-inflammatory cytokines secretion in Caco-2 cell line in vitro. The subject is very interesting and worth of studying. However, positive controls have been missed in both models.

The title should be changed, e.g. “A role of exopolysaccharide produced by Streptococcus thermophilus in the intestinal inflammation and protection of mucosal barrier in Caco-2 monolayer and experimental murine colitis”

More details on the potential probiotic role of Streptococcus thermophilus should be provided in the introduction. Otherwise the question is why the Authors chose this strain to gain EPS-1.

L. 80 Please explain GPC when it is used for the first time. Additionally, please check other abbreviations AJ and DAI if they are used for the first time in the text.

In the materials and methods with Caco-2 monolayer,  please include the pore density of membranes and the material which they were made of for TEER and transport studies. Could you provide the optimal TEER of monolayer considered and used in the experiments.

Why was EPS-1 applied into the apical and basal chamber? It is more reasonable to apply it only in the apical chamber of Transwell.

L. 210. It should be “A previous study…”

L. 247-248 Please check the font.

Thus, the manuscript is recommended for publication after minor revision.

Author Response

Comments and Suggestions for Authors

The manuscript (Ref. no.molecules-425482) have described the effects of exopolysaccharide isolated from Streptococcus thermophilus on the DSS-induced colitis and tight junction disruption  in mice as well as pro-inflammatory cytokines secretion in Caco-2 cell line in vitro. The subject is very interesting and worth of studying. However, positive controls have been missed in both models.

(1)The title should be changed, e.g. “A role of exopolysaccharide produced by Streptococcus thermophilus in the intestinal inflammation and protection of mucosal barrier in Caco-2 monolayer and experimental murine colitis”

Response 1:

Thank you for your suggestions. We have adopted this title in revised manuscript as your opinion.

(2)More details on the potential probiotic role of Streptococcus thermophilus should be provided in the introduction. Otherwise the question is why the Authors chose this strain to gain EPS-1.

Response 2:

Thank you for your suggestions. We have added more information (maximum yield of EPS and genomic sequence) about. S. thermophilus MN-BM-A01

(3)L. 80 Please explain GPC when it is used for the first time. Additionally, please check other abbreviations AJ and DAI if they are used for the first time in the text.

Response 3:

We are sorry for the careless and these mistakes. We have corrected these mistakes in revised manuscript.

(4)In the materials and methods with Caco-2 monolayer,  please include the pore density of membranes and the material which they were made of for TEER and transport studies. Could you provide the optimal TEER of monolayer considered and used in the experiments.

Response 4:

We are sorry for the unclear description. Caco-2 cells were cultured in DMEM and seeded at a density of 1 x 105 cells per upper well. After culturing, Caco-2 cells become a full monolayer with a mean TEER exceeding 200 Ω·cm2. Keep on culturing until the TEER exceeded 400 Ω·cm2 measured by a Millicell-Electrical Resistance System voltohmmeter and the TEER kept stable for 3 days. At that time, we thought a confluent monolayer was obtained and the Caco-2 cells had been fully differentiation. Then it could be used for the study in our experiment. We have added these information in revised manuscript.

(5)Why was EPS-1 applied into the apical and basal chamber? It is more reasonable to apply it only in the apical chamber of Transwell.

Response 5:

We are sorry for the careless and these mistakes. EPS-1 (2mg/mL) was added to the apical chamber of Transwell supports. We have corrected this mistake in revised manuscript.

(6)L. 210. It should be “A previous study…”

Response 6:

We are sorry for the careless and these mistakes. We have corrected the mistake in revised manuscript.

(7)L. 247-248 Please check the font.

Response 7:

We are sorry for the careless and these mistakes. We have corrected these mistakes in revised manuscript.

Reviewer 3 Report

A water-soluble EPS-1 isolated from MN-BM-A01 was composed of rhamnose, glucose and galactose in a molar ratio of 2.6:1.3:6.1, with molecular weight of 4.23×106 Da. After EPS-1 administration, the disease severity of mouse colitis was significantly alleviated, mainly manifesting as the decrease of disease activity index and mitigated colonic epithelial cell injury. Meanwhile, proinflammatory cytokines levels were significantly suppressed, the reduced expressions of tight junction protein were counteracted. In addition, the results in vitro showed that EPS-1 protected intestinal barrier integrity from the disruption by LPS in Caco-2 monolayer, increased expression tight junction and alleviated proinflammatory response. The study confirmed the protective effects of purified EPS produced by Streptococcus thermophilus on acute colitis via alleviating intestinal inflammation and improving mucosal barrier function. This manuscript is well organized but some minor points should be checked before publishing.

Major points:

The figure number indicated in text should be rearranged (i.e. Fig 2 to 5 was indicated as Fig 1 to 4 in the text, respectively).

Minor points:

1.      Line 22: “TNF-α, IL-6 and IFN-γ” should be “tumor necrosis factor-α, interlekin-6 and interferon-g”.

2.      Line 25: “LPS” should be “lipopolysaccharide”.

3.      Line 38: “found” should be “finding”.

4.      Line 51: “MLCK” should be “myosin light chain kinase (MLCK)”.

5.      Line 52: “IL-13” should be “interleukin (IL)-13”.

6.      Line 80: “GPC” should be “gel-permeation chromatography (GPC)”.

7.      Line 100: “DAI” should be “disease activity index (DAI)”.

8.      Line 123: “TNF-α, IL-6 and IFN-γ” should be tumor necrosis factor-α (TNF-α), IL-6 and interferon-g (IFN-γ)”.

9.      Line 149: “TER” should be “transepithelial electrical resistance (TER)”.

10.  Line 150: “LPS” should be “lipopolysaccharide (LPS)”.

11.  Line 152: “FITC” should be “fluorescein isothiocyanate (FITC)”.

12.  Line 225: “IBD” should be “inflammatory bowl disease (IBD)”.

13.  Line 264: “Superdex75” should be “Superdex 75”.

14.  Line 318: “ELISA” should be “The enzyme-linked immunosorbent assay (ELISA)”.

15.  Line 332: “SPSS” should be “Statistical Product and Service Solutions (SPSS)”.

16.  Line 334: “ANOVA” should be “analysis of variance (ANOVA)”.

Author Response

Response to Reviewer 3 Comments

Comments and Suggestions for Authors

Major points:

The figure number indicated in text should be rearranged (i.e. Fig 2 to 5 was indicated as Fig 1 to 4 in the text, respectively).

 Response :

We are sorry for the careless and these mistakes. We have corrected these mistakes in revised manuscript.

Minor points:

1.      Line 22: “TNF-α, IL-6 and IFN-γ” should be “tumor necrosis factor-α, interlekin-6 and interferon-g”.

Response :

We are sorry for the careless and these mistakes. We have corrected these mistakes in revised manuscript.

2.      Line 25: “LPS” should be “lipopolysaccharide”.

Response :

We are sorry for the careless and these mistakes. We have corrected this mistake in revised manuscript.

3.      Line 38: “found” should be “finding”.

Response :

We are sorry for the careless and these mistakes. We have corrected this mistake in revised manuscript.

4.      Line 51: “MLCK” should be “myosin light chain kinase (MLCK)”.

Response :

We are sorry for the careless and these mistakes. We have corrected this mistake in revised manuscript.

5.      Line 52: “IL-13” should be “interleukin (IL)-13”.

Response :

We are sorry for the careless and these mistakes. We have corrected this mistake in revised manuscript.

6.      Line 80: “GPC” should be “gel-permeation chromatography (GPC)”.

Response :

We are sorry for the careless and these mistakes. We have corrected this mistake in revised manuscript.

7.      Line 100: “DAI” should be “disease activity index (DAI)”.

Response :

We are sorry for the careless and these mistakes. We have corrected this mistake in revised manuscript.

8.      Line 123: “TNF-α, IL-6 and IFN-γ” should be “tumor necrosis factor-α (TNF-α), IL-6 and interferon-g (IFN-γ)”.

Response :

We are sorry for the careless and these mistakes. We have corrected these mistakes in revised manuscript.

9.      Line 149: “TER” should be “transepithelial electrical resistance (TER)”.

Response :

We are sorry for the careless and these mistakes. We have corrected this mistake in revised manuscript.

10.  Line 150: “LPS” should be “lipopolysaccharide (LPS)”.

Response :

We are sorry for the careless and these mistakes. We have corrected this mistake in revised manuscript.

11.  Line 152: “FITC” should be “fluorescein isothiocyanate (FITC)”.

Response :

We are sorry for the careless and these mistakes. We have corrected this mistake in revised manuscript.

12.  Line 225: “IBD” should be “inflammatory bowl disease (IBD)”.

Response :

We are sorry for the careless and these mistakes. We have corrected this mistake in revised manuscript.

13.  Line 264: “Superdex75” should be “Superdex 75”.

Response :

We are sorry for the careless and these mistakes. We have corrected this mistake in revised manuscript.

14.  Line 318: “ELISA” should be “The enzyme-linked immunosorbent assay (ELISA)”.

Response :

We are sorry for the careless and these mistakes. We have corrected this mistake in revised manuscript.

15.  Line 332: “SPSS” should be “Statistical Product and Service Solutions (SPSS)”.

Response :

We are sorry for the careless and these mistakes. We have corrected this mistake in revised manuscript.

16.  Line 334: “ANOVA” should be “analysis of variance (ANOVA)”.

 Response :

We are sorry for the careless and these mistakes. We have corrected this mistake in revised manuscript.

Round 2

Reviewer 1 Report

Response to Reviewer 1 Comments

General comments:

1) Polysaccharides are rapidly metabolized by the organism. It is probably unlikely that they may have any effects as anti-inflammatory agents and on improving mucosal barrier function. To fully address the putative positive effects of polysaccharides, diet containing only monosaccharides such as rhamose, glucose and galactose shall have been performed.

Response 1:

Thank you for your suggestions. Exopolysaccharides (EPS) are polymers synthesised by a range of bacterial groups. In general, EPS was not easy to digest which was shown to protect the bacterial cell from environmental stresses, such as human gastric and pancreatic enzymes, bile salts and varying pH [1]. A previous report has improved that EPS produced by Lactococcus lactis ssp. cremoris B40, Lactobacillus sakei 0–1, Streptococcus thermophilus SFi20, and Lactobacillus helveticus Lh59 all retain integrity during gastric transit [2].

For this reason, the undigested EPS can reach the lower intestine and positively impact on the gut microbiome and indirectly modulate the immune system. It was well documented that dendritic cells (DC) sample luminal microbes and microbial components as antigens, such as EPS, resulting in augmentation of natural killer cell activity and modulate inflammatory cytokine expression including interleukins, interferons and tumour necrosis factor [3, 4]. In this study, anti-inflammatory effect of EPS may be related to this mechanism and we will study further next.

Reference:

1.         Ryan, P. M.; Ross, R. P.; Fitzgerald, G. F.; Caplice, N. M.; Stanton, C., Sugar-coated: exopolysaccharide producing lactic acid bacteria for food and human health applications. Food Funct 2015, 6, (3), 679-693.

2.         Ruijssenaars, H. J.; Stingele, F.; Hartmans, S., Biodegradability of food-associated extracellular polysaccharides. Curr Microbiol 2000, 40, (3), 194-199.

3.         Farache, J.; Koren, I.; Milo, I.; Gurevich, I.; Kim, K. W.; Zigmond, E.; Furtado, G. C.; Lira, S. A.; Shakhar, G., Luminal Bacteria Recruit CD103(+) Dendritic Cells into the Intestinal Epithelium to Sample Bacterial Antigens for Presentation. Immunity 2013, 38, (3), 581-595.

4.         Arena, A.; Maugeri, T. L.; Pavone, B.; Iannello, D.; Gugliandolo, C.; Bisignano, G., Antiviral and immunoregulatory effect of a novel exopolysaccharide from a marine thermotolerant Bacillus licheniformis. International immunopharmacology 2006, 6, (1), 8-13. 

Answer to authors: This concern was not adequately addressed. There was no experimental evidence that exopolysaccharides remained undigested in mice. It is unclear how such large molecules can interact specifically with tissues and/or endothelial cells. The exopolysaccharides can be digested by microbiome gut. Therefore diet containing rhamose, glucose and galactose shall have been performed to determine possible effects of monosaccharides.

2) It is not possible to rule out that other compounds than polysaccharides could have possible beneficial effects. In this respect, sample-to-sample variation was not adequately addressed. The mass spectra was not fully exploited to pint point possible other compounds.

Response 2:

We are sorry for the unclear description. Just as your concern, the purity of EPS is very important for our study. The EPS was isolated and purified as previously described [5]. There were four steps in our study to ensure the purity of EPS. Firstly, after fermentation, trichloroacetic acid (TCA) was added to the culture to a final concentration of 4% (w/v), and the mixture was stirred for 30 min at room temperature. Cells and precipitated proteins were removed by centrifugation (8,000 × g, 4 ◦C, 10 min). Secondly, crude EPS was precipitated from the supernatant by addition of 2 volumes of cold ethanol stored at 4◦C for 24 h. Crude EPS was collected by centrifugation at 8,000 × g for 10 min. Most of the rest proteins, liposoluble and other components were removed. Thirdly, crude EPS solution was dialyzed using a dialysis bag (Mw cut-off 8 kDa to 14 kDa) against distilled water for 48 h at 4 °C to remove the low molecular weight components. Fourthly, before the crude EPS solution was fractionated with an anion exchange chromatography, we tested the sample again for protein, amino acid and polypeptide content to rule out interference from other compounds. We have revised the original manuscript in this part to make it easier for readers to understand.

The average molecular weights of the purified EPS were measured by gel permeation chromatography (GPC) rather than mass spectroscopy. GPC is a commonly used method for determining the average molecular weight of polysaccharides [5, 6]. Standard dextrans (4, 10, 32, 100, 500kDa, Fluka Chemical Co., Buchs, Switzerland) were passed through Waters Ultra-hydrogel 250, 1000 and 2000 (7.8 × 300 mm) columns in series. The column was eluted with 20 mM CH3COONH4 solution at a flow rate of 0.5 mL/min, and the injection volume of sample was 20 μL at an internal temperature of 40 °C. The standard curve equation Log Mw = -0.1741x + 11.505 (R2 = 0.9913), where Mw is the peak molecular weight and x is the retention time (figure 1 below).

Figure 1 Standard curve of dextrans measured by gel permeation chromatography

Thank you for your suggestions. In the revised manuscript, molar ratio of rhamnose, glucose, galactose and mannose was changed to 12.9: 26.0: 60.8: 0.25. We analyzed the three extracted EPS samples. The approximate molar ratio of the sugar monomers was shown in revised figure 1C and the table below. The relative proportion of four different sugar monomers varied slightly between samples. The proportion of mannose was very low (0.25%).

Sample NO.

Rhamnose(%)

Glucose(%)

Galactose(%)

Mannose(%)

1

12.54

25.81

61.40

0.25

2

13.43

26.14

60.20

0.23

3

12.68

25.97

61.08

0.27

Average proportion

12.88

25.97

60.8%

0.25

Standard deviation

0.48

0.16

0.62

0.02

Reference:

5.         Zhang, J.; Cao, Y.; Wang, J.; Guo, X.; Zheng, Y.; Zhao, W.; Mei, X.; Guo, T.; Yang, Z., Physicochemical characteristics and bioactivities of the exopolysaccharide and its sulphated polymer from Streptococcus thermophilus GST-6. Carbohydrate polymers 2016, 146, 368-75.

Answer to authors: It was not adequately addressed. The authors give convincible information on composition and purity of polysaccharides. However, it is not clear how undigested exopolysaccharides can interact with heterogeneous human epithelial colorectal adenocarcinoma cells (Caco-2 cell line). A careful inspection of alternate compounds is needed to fully convince that an intact large polysaccharide can interact with human epithelial cells.

3) The molecular mechanisms of polysaccharides and of monosaccharides to trigger anti-inflammatory response or improving mucosal barrier function need to be tackled to fully convince that such molecules could have any biological effects.

Response 3:

Thank you for your suggestions. As discussed above, EPS was not easy to digest, the undigested intact EPS can reach the intestine and positively impact on the gut microbiome and indirectly modulate the immune system. Dendritic cells (DC) sampled EPS as antigens, resulting in immunoregulation of inflammatory cytokine expression including interleukins, interferons and tumour necrosis factor [3, 4]. In this study, anti-inflammatory effect of EPS maybe related to this mechanism and we will study further next.

Reference:

3.         Farache, J.; Koren, I.; Milo, I.; Gurevich, I.; Kim, K. W.; Zigmond, E.; Furtado, G. C.; Lira, S. A.; Shakhar, G., Luminal Bacteria Recruit CD103(+) Dendritic Cells into the Intestinal Epithelium to Sample Bacterial Antigens for Presentation. Immunity 2013, 38, (3), 581-595.

4.         Arena, A.; Maugeri, T. L.; Pavone, B.; Iannello, D.; Gugliandolo, C.; Bisignano, G., Antiviral and immunoregulatory effect of a novel exopolysaccharide from a marine thermotolerant Bacillus licheniformis. International immunopharmacology 2006, 6, (1), 8-13. 

Answer to authors: It was not adequately addressed. As the authors pointed out intact polysaccharide shall have an impact on gut microbiome. In this report there were no findings on the effects of intact exopolysaccharide to gut microbiome.

4) The coverage of literature was not critically assessed and some paragraphs lacked appropriate references.

Response 4:

We are sorry for the careless and these mistakes. We have corrected these mistakes in revised manuscript.

Answer to authors: It was not adequately addressed. The relationship between gut microbiome and epithelial cells was not sufficiently explained. A more critical assessment of literature shall be performed especially that there are confusions on the true target of exopolysaccharides and mechanisms of inflammation.

5) Ethical concerns for animal uses were not addressed.

Response 5:

 Thank you for your suggestions. Ethical concerns for animal was shown below.

Answer to authors: It was not addressed. What is needed is a statement in “4.4 Animals, diets and experimental 281 procedure” where ethical concerns are addressed, as in the case for all experiments on animals. There is no point to provide a copy of a form that is not included in the manuscript.

Minor comments:

6) Title: Change title since there is abbreviation DSS in the title.

Response We are very sorry for the mistakes and we have corrected the text accordingly.

Answer to authors: It was addressed.

7) Abstract: Delete abbreviation UC in the abstract since it was mentioned only once in the abstract.

Response We are very sorry for the mistakes and we have corrected the text accordingly.

Answer to authors: It was addressed.

8) Abstract: Delete abbreviation LPS and specify LPS

Response: We are very sorry for the mistakes and we have corrected the text accordingly.

Answer to authors: It was addressed.

9) Abstract: Delete abbreviations TNF-α, IL-6 and IFN-γ and specify.

Response: We are very sorry for the mistakes and we have corrected the text accordingly.

Answer to authors: It was addressed.

10) Abstract: Specify Caco-2

Response: Thank you for your suggestions. But as far as I know, the Caco-2 cell line is a continuous line of human epithelial colorectal adenocarcinoma cells. “Caco-2” is not an abbreviated form.

Answer to authors: The authors shall add “continuous line of human epithelial colorectal adenocarcinoma” to specify Caco-2 cells.

11) Introduction lines 37-38: The sentence is unclear:” which implies that lack of effective strategy was the determining factor for UC patients”.

Response: We are sorry and we have modified sentence to make it clear.

Answer to authors: It was addressed.

12) Introduction, lines 39-41: Provide a reference to support the sentence “Although, precise mechanism of UC is still unclear, while there is no doubt 39 that intestinal mucosal barrier dysregulation and increased paracellular permeability play a critical role in pathogenesis of UC.”

Response: Thank you for your advice. We have added a new reference to support the sentence.

Answer to authors: It was addressed.

13) Introduction, lines 41-42: Provide a reference to support “Tight junctions (TJ) of enterocyte regulates the integrity of intestinal barrier predominantly, which influence the paracellular and transcellular transport.”

Response: Thank you for your advice. We have added a new reference to support the sentence.

Answer to authors: It was addressed.

14)Introduction lines 44-46: Provide several references to support “Many researches pointed out that, in UC patients, downregulations of tight junctions giving rise to the invasion of the colon wall with intestinal pathogens and toxins, which was considered as a vital event in the pathogenesis of intestinal inflammation.”

Response: Thank you for your advice. We have added references to support the sentence.

Answer to authors: It was addressed.

15) Introduction line 51: Specify abbreviation MLCK.

Response: We are very sorry for the mistakes and we have corrected the text accordingly.

Answer to authors: It was addressed.

16) Introduction line 52: Specify abbreviation IL.

Response: We are very sorry for the mistakes and we have corrected the text accordingly. Answer to authors: It was addressed.

17) Introduction, lines 55-57: Provide references to support “Recently, increasingly evidences supported that probiotic supply beneficial effects for host health, especially in terms of intestinal function improvement and prevention of a variety of intestinal diseases including UC.”

Response: Thank you for your advice. We have added references to support the sentence.

Answer to authors: It was addressed.

18) Introduction, line 67: Specify MN-BM-A01 (CGMCC No. 11383).

Response: Thank you for your suggestions. However, MN-BM-A01 is the full name of the strain rather than the abbreviation. “CGMCC No. 11383” is the strain preservation number in China general microbiological culture collection center.

Answer to authors: It was not addressed. It shall be stated that MN-BM-A01 (CGMCC No. 11383) is a Streptococcus thermophilus strain isolated from Yogurt Block in Gansu, China.

19) Results, line 82: Provide mass determination by mass spectroscopy to confirm chromatography data and to ascertain distribution of molecular mass.

Response: The average molecular weights of the purified EPS were measured by gel permeation chromatography (GPC) rather than mass spectroscopy. GPC is a commonly used method for determining the average molecular weight of polysaccharides [5, 6]. Standard dextrans (4, 10, 32, 100, 500kDa, Fluka Chemical Co., Buchs, Switzerland) were passed through Waters Ultra-hydrogel 250, 1000 and 2000 (7.8 × 300 mm) columns in series. The column was eluted with 20 mM CH3COONH4 solution at a flow rate of 0.5 mL/min, and the injection volume of sample was 20 μL at an internal temperature of 40 °C. The standard curve equation Log Mw = -0.1741x + 11.505 (R2 = 0.9913), where Mw is the peak molecular weight and x is the retention time (figure 1 below).

Figure 1 standard curve of dextrans measured by gel permeation chromatography

Reference:

5.         Zhang, J.; Cao, Y.; Wang, J.; Guo, X.; Zheng, Y.; Zhao, W.; Mei, X.; Guo, T.; Yang, Z., Physicochemical characteristics and bioactivities of the exopolysaccharide and its sulphated polymer from Streptococcus thermophilus GST-6. Carbohydrate polymers 2016, 146, 368-75.

6.         Striegel, A. M.; Timpa, J. D., Molecular Characterization of Polysaccharides Dissolved in Me(2)Nac-Licl by Gel-Permeation Chromatography. Carbohyd Res 1995, 267, (2), 271-290.

Answer to authors: It was addressed.

20 ) Results line 86: Chromatography profiles indicated the presence of fucose, and mannose in addition to rhamose, glucose and galactose that were not reported in the text.

Response: We are sorry for the unclear description and thanks for your suggestions. We added the proportion of mannose and provided a new chromatography profile (fig 1C) in revised manuscript. EPS-1 was composed of four different sugar monomers including rhamnose, glucose, galactose and mannose. We analyzed the three extracted EPS samples. The approximate molar ratio of the sugar monomers was shown in table below. The relative proportion of four different sugar monomers varied slightly between samples. The proportion of mannose was very low (0.25%).

Sample NO.

Rhamnose(%)

Glucose(%)

Galactose(%)

Mannose(%)

1

12.54

25.81

61.40

0.25

2

13.43

26.14

60.20

0.23

3

12.68

25.97

61.08

0.27

Average proportion

12.88

25.97

60.8%

0.25

Standard deviation

0.48

0.16

0.62

0.02

Answer to authors: It was addressed. However, this table shall be put in the manuscript.

21) Results line 87: There was two chromatograms in figure 1C that were not adequately described in figure legend.

Response: We are sorry for the unclear description. We provided chromatography profile (fig 1C) and modified the figure legend in revised manuscript.

Answer to authors: It was addressed.

22) Results line 86-87: Provide sample to sample errors on composition of monosaccharides.

Response: We analyzed the three extracted EPS samples. The approximate molar ratio of the sugar monomers was shown in table below.

Sample NO.

Rhamnose(%)

Glucose(%)

Galactose(%)

Mannose(%)

1

12.54

25.81

61.40

0.25

2

13.43

26.14

60.20

0.23

3

12.68

25.97

61.08

0.27

Average proportion

12.88

25.97

60.8%

0.25

Standard deviation

0.48

0.16

0.62

0.02

 Answer to authors: It was addressed. However, this table shall be put in the manuscript.

23) Results line 107: Specify abbreviation HE.

Response: We are very sorry for the mistakes and we have corrected the text accordingly.

Answer to authors: It was addressed.

24) Results line 112 and Fig 2E: It is unclear when histological scores were determined and so far there were no findings suggesting an increase of histological scores at day 14.

Response: We are sorry for the unclear description. At the end of the experiment (day 14), after sacrificed, histological injury score of each colon was evaluated by morphological criteria as the protocol previously described. The length of colons was measured and cut open longitudinally 1 cm of the distal colon for formalin fixing, paraffin embedding and HE staining. The paraffin sections of colon were graded by two blinded investigators with a range from 0 to 3 as to amount of inflammation (acute and chronic), depth of inflammation and with a range from 0 to 4 as to the amount of crypt damage or regeneration as previously described [7]. Histological scores was the sum of these scores.

Reference:

7.    Dieleman, L. A.; Palmen, M. J.; Akol, H.; Bloemena, E.; Pena, A. S.; Meuwissen, S. G.; Van Rees, E. P., Chronic experimental colitis induced by dextran sulphate sodium (DSS) is characterized by Th1 and Th2 cytokines. Clinical and experimental immunology 1998, 114, (3), 385-91.

Answer to authors: It was addressed. However, I suggest to change the sentence from “Histological scores were increased at day 14 in DSS-treated group compared to the control group” to “At the end of experiment (day 14), histological scores were determined. The histological scores in DSS-treated group increased compared to the control group.” The confusion arises from the wrong perception that there were no changes before day 14, while the authors confirmed there were no measurements performed before that time. Therefore the sentence needs to be clarified.

25) Results line 124: Change Figure 2A by Figure 3A

Response: We are very sorry for the mistakes and we have corrected the text accordingly.

Answer to authors: It was addressed.

26) Results line 129: Change Figure 2B by Figure 3B.

Response: We are very sorry for the mistakes and we have corrected the text accordingly.

Answer to authors: It was addressed.

27) Results, lines 133 and 136: Change Figure 3 by Figure 4.

Response: We are very sorry for the mistakes and we have corrected the text accordingly.

Answer to authors: It was addressed.

28) Results, line 133 and 136: Specify Figure 4A and Figure 4B.

Response: We are very sorry for the mistakes and we have corrected the text accordingly.

Answer to authors: It was addressed.

29) Results, lines 150, 155, 156 and 162: Change Figure 4 by Figure 5.

Response: We are very sorry for the mistakes and we have corrected the text accordingly.

Answer to authors: It was addressed.

30) Results, lines 150-165: There was no control on the viability of cells and it is unclear if the changes were related to toxicity of the treatment.

Response: We are sorry for the unclear description. In cellular experiments, EPS (2mg/mL) treatment alone was regarded as control to measure the toxicity of EPS.

Answer to authors: It is not addressed since the same experiment without EPS-1 shall have been performed as stated in the text “without or with EPS-1 for 24 hours” (line 157) which contradicts the lack of findings (Fig 5A) on treatment without EPS-1 with or without LPS. Besides it is not clear what is the control.

31) Results, line 156, Figure 5D: Provide quantitative estimation of Western Blot profiles.

Response: Thank you for your advice. We have added a new fig.5E to show the quantitative analysis of Western Blot profiles

Answer to authors: It was addressed. However, provide statistical significances in Fig 5E to substantiate significant changes in protein expressions.

32) Results, line 162, Figure 5C: Indicate on the ordinate what correspond the numbers.

Response: We are very sorry for the mistakes and we have added the ordinate in fig. 5C.

Answer to authors: It was addressed.

33) Discussion, lines 175-176: Provide one reference to support that “Ulcerative colitis (UC) is a set of complicated chronic inflammatory and ulceration conditions of the colonic mucosa, accompanied by clinical symptoms.”

Response: Thank you for your advice. We have added a new reference to support the sentence.

Answer to authors: It was addressed.

33) Discussion, lines 176-177: Provide references to support that “Recently, increasingly evidences supported that probiotics conferred a lot of health benefits, including UC prevention”

Response: Thank you for your advice. We have added new references to support the sentence.

Answer to authors: It was addressed.

34) Discussion, lines 176-177: Provide references to support that “EPS may play a role in this prevention process. However, there is no literature reporting on the intervention effects of purified EPS in experimental colitis. ”

Response: Thank you for your advice. We have added new references to support the sentence.

Answer to authors: It was addressed.

35) Discussion, lines 185-186: Provide references to support that S. thermophilus is the most important dairy starter. The exopolysaccharides of S. thermophilus can improve the properties of the dairy product and confer benefcial health effects.

Response: Thank you for your advice. We have added new references to support the sentence.

Answer to authors: It was addressed.

36) Discussion, lines 190-192: At this stage no firm conclusion can be drawn on the composition of exopolysaccharide due to lack of sample to sample determinations and beside two monosacharrides fucose and mannose were omitted (Figure 1C) in the composition. The sentence “In this study, the EPS-1 produced by S. thermophilus MN-BM-A01 was a representative heteropolysaccharide, consisting of rhamnose, glucose and galactose in a molar ratio of 2.6: 1.3: 6.1.” shall be deleted.

Response: We are sorry for the unclear description and thanks for your suggestions. We added the proportion of mannose and provided a new chromatography profile (fig 1C) in revised manuscript. We analyzed the three extracted EPS samples. The approximate molar ratio of the sugar monomers was shown in table below. In summary, EPS was composed of different sugar monomers including rhamnose, glucose, galactose and mannose in an approximate molar ratio of 12.9 : 26.0 : 60.8 : 0.25

Sample NO.

Rhamnose(%)

Glucose(%)

Galactose(%)

Mannose(%)

1

12.54

25.81

61.40

0.25

2

13.43

26.14

60.20

0.23

3

12.68

25.97

61.08

0.27

Average proportion

12.88

25.97

60.8%

0.25

Standard deviation

0.48

0.16

0.62

0.02

Answer to authors: It was addressed.

However this table shall be put in the manuscript (see comment 20 and 22).

37) Discussion, line 193: The sentence” UC is a chronic inflammatory disease of large intestine.” Shall be deleted, it is stated in lines 175-176.

Response: We are very sorry for the mistakes and we have corrected the text accordingly.

Answer to authors: It was addressed.

38) Discussion, lines 193,194: Provide one reference to support that “The occurrence and development of UC were strongly associated with the imbalance of pro-inflammatory and anti-inflammatory cytokines”.

Response: Thank you for your advice. We have added a new reference to support the sentence.

Answer to authors: It was addressed.

39) Discussion, lines 195,196: Provide references to support that “Previous reports showed that a UC-like mouse model of DSS-induced colitis had neutrophil accumulation and increased expression of pro-inflammatory in the colon.”

Response: Thank you for your advice. We have added new references to support the sentence.

Answer to authors: It was addressed.

40) Discussion, lines 195,194: There were no experimental evidence that cytokines were released by macrophages. The sentence “Our study also demonstrated that production of macrophage-derived cytokines TNF-α, IFN-γ and IL-6” shall be modified or deleted.

Response: We are very sorry for the mistakes and we have corrected the text accordingly.

Answer to authors: It was addressed.

41) Discussion, lines 204-206: So far the experiments were performed on mice and not on rats: The sentence” Our study also demonstrated that production of macrophage-derived cytokines TNF-α, IFN-γ and IL-6 increased significantly, but anti-inflammatory cytokines (IL-4 and IL-10)” shall be deleted or modified.

Discussion, line 208-213: Specify probiotic in several sentences “Although therapeutic effects of probiotics on colitis were strain-specific and dose-dependant, the anti-inflammatory effects of probiotics play a decisive role [27, 28].P revious study has indicated that the expression of TNF-α, IL-1β, and IL-6 in the colon tissues was diminished dose-dependently by a mixture of three potential probiotic strains in DSS induced colitis[29]. Pan T. et.al also illustrated that an appropriate dose of probiotics strain can prevent intestinal damage in mice with DSS-induced colitis by inhibiting the expression of inflammatory cytokines [30].”

Response: We are very sorry for the mistakes and we have corrected the text accordingly.

Answer to authors: It was addressed.

42) Discussion, lines 213-218: The sentences “In our study, EPS-1 treatment attenuated the release of TNF-α, IFN-γ and IL-6 significantly, suggesting that this effect is the primary cause of anti-colitic activity of EPS. The inhibition effect or pro-inflammatory cytokines was further confirmed in Caco-2 cell lines treated by LPS with or without EPS-1. Results demonstrated that EPS-1 attenuated the LPS-induced inflammatory response of the mucosa and the TER reduction,accompanying significantly alleviated morphological disruption of tight junction protein.” Shall be softened since it is unlikely that polysaccharides could have an anti-inflammatory effect, since they are rapidly metabolized.

Response:

Thank you for your suggestions. Exopolysaccharides (EPS) are polymers synthesised by a range of bacterial groups. In general, EPS was not easy to digest which was shown to protect the bacterial cell from environmental stresses, such as human gastric and pancreatic enzymes, bile salts and varying pH [1]. A previous report has improved that EPS produced by Lactococcus lactis ssp. cremoris B40, Lactobacillus sakei 0–1, Streptococcus thermophilus SFi20, and Lactobacillus helveticus Lh59 all retain integrity during gastric transit [2].

For this reason, the undigested EPS can reach the lower intestine and positively impact on the gut microbiome and indirectly modulate the immune system. It was well documented that dendritic cells (DC) sample luminal microbes and microbial components as antigens, such as EPS, resulting in augmentation of natural killer cell activity and modulate inflammatory cytokine expression including interleukins, interferons and tumour necrosis factor [3, 4]. In this study, anti-inflammatory effect of EPS may be related to this mechanism and we will study further next.

Reference:

1.         Ryan, P. M.; Ross, R. P.; Fitzgerald, G. F.; Caplice, N. M.; Stanton, C., Sugar-coated: exopolysaccharide producing lactic acid bacteria for food and human health applications. Food Funct 2015, 6, (3), 679-693.

2.         Ruijssenaars, H. J.; Stingele, F.; Hartmans, S., Biodegradability of food-associated extracellular polysaccharides. Curr Microbiol 2000, 40, (3), 194-199.

3.         Farache, J.; Koren, I.; Milo, I.; Gurevich, I.; Kim, K. W.; Zigmond, E.; Furtado, G. C.; Lira, S. A.; Shakhar, G., Luminal Bacteria Recruit CD103(+) Dendritic Cells into the Intestinal Epithelium to Sample Bacterial Antigens for Presentation. Immunity 2013, 38, (3), 581-595.

4.         Arena, A.; Maugeri, T. L.; Pavone, B.; Iannello, D.; Gugliandolo, C.; Bisignano, G., Antiviral and immunoregulatory effect of a novel exopolysaccharide from a marine thermotolerant Bacillus licheniformis. International immunopharmacology 2006, 6, (1), 8-13.

Answer to authors: It was not adequately addressed. The sentence needs to be softened to include possible unidentified compounds that may induce anti-inflammation response. I am not convinced that EPS can trigger itself anti-inflammation response by Caco-2 cells. In my opinion it is unlikely to have interaction of intact EPS to endothelial cells triggering anti-inflammatory response.

43) Discussion lines 237-238: The sentence “Our data also revealed that EPS-1 facilitated the gut barrier function in vitro and in vivo and increased the expression of tight junction proteins, which might be caused by the suppressing of pro-inflammatory cytokines” shall be softened.

Response: Thank you for your suggestions. We have made corresponding modifications according to your opinions.

Answer to authors: It was addressed.

44) Conclusion, lines 242-248: The whole paragraph “In this study, a water-soluble exopolysaccharide, designated as EPS-1, was isolated from MN-BM-A01 and purified. EPS-1 was a heteropolysaccharide, which was composed of rhamnose, glucose and galactose in a molar ratio of 2.6: 1.3: 6.1. Our study confirmed the protective effects of purified EPS-1 on acute mouse colitis, mainly manifesting as the decrease of DAI and mitigated colonic epithelial cell injury. The protective effects might be related to the alleviation of intestinal inflammation and the improvement of mucosal barrier function. Thus, EPS-1 may be a preventive therapeutic agent for UC and a new health care product for intestinal health.” Is already stated in the discussion and shall be deleted.

Response: Thank you for your suggestions. We have made corresponding modifications according to your opinions.

Answer to authors: It was addressed.

45) Materials and Methods, lines  251, 253, 256, 285, 305, 306 and 326: Specify units in %.

Response: We are very sorry for the mistakes and we have corrected the text accordingly.

Answer to authors: It was not adequately addressed:  lines 251, line 253, line 257, 288, 310, 344. Replace w:v by units. Indeed, it is not clear in which units are w:v, it could be g:mL or mg:mL.

46) Materials and Methods, line 254: Indicate in which buffer (composition and final pH) was added trifluoroacetic acid.

Response: We are sorry for the unclear description. Trichloroacetic acid was added to the culture to a final concentration of 4% (w/v), and the mixture was stirred for 30 min at room temperature. Cells and precipitated proteins were removed by centrifugation. There was no buffer added to the culture.

Answer to authors: It was addressed.

47) Materials and methods, line 263: Provide more information on further purification.

Response: We are sorry for the unclear description, and we have modified section to make it clearly. The purified EPS-1 solution (10 g/mL, 10 mL) was performed by an automatic polysaccharide gel purification system (Superdex75) with column (1.6 × 80 cm), eluted with 0.5 % NaCl solution at a flow rate of 0.2 mL/min. Every 2 mL of elution was collected automatically and the carbohydrate content was determined by phenol–sulfuric acid method. Peak fractions containing polysaccharides were pooled, dialyzed, and lyophilized.

Answer to authors: It was addressed.

48) Materials and Methods, line 269 replace CH3COONH4 by CH3COONH4

Response: We are very sorry for the mistakes and we have corrected the text accordingly.

Answer to authors: It was addressed.

49) Materials and Methods, line 272: Specify TFA.

Response: We are very sorry for the mistakes and we have corrected the text accordingly.

Answer to authors: It was addressed.

50) Materials and Methods, lines 279: Provide information on ethical committee.

Response: Thank you for your suggestions. Ethical concerns for animal was shown below.

Answer to authors: It was not addressed. What is needed is a statement in “4.4 Animals, diets and experimental 281 procedure” where ethical concerns are addressed, as in the case for all experiments on animals. There is no point to provide a copy of a form that is not included in the manuscript.

51) Materials and Methods, line 279: Specify BALB/c.

Response: Thank you for your suggestions. But as far as I know, “BALB/c” is not an abbreviated form.In 1974, this substrain of mouse were returned to The Jackson Laboratory and were named BALB/c. BALB/c mice are distributed globally, and are among the most widely used inbred strains used in animal experimentation.

Reference:

A Brief History of the Two Substrains of BALB/c, BALB/cJ, and BALB/cByJ". Jax Mice Literature. Jackson Laboratory. Archived from the original on 19 June 2011. 2010-09-30.

Answer to authors: It was addressed.

52) Materials and Methods, line 295: Specify when the length of colons was measured.

Response: Thank you for your advice. At the end of the experiment (day 14), after sacrificed , the length of colons was measured, and we have corrected the text accordingly.

Answer to authors: It was addressed.

53) Materials and Methods, line 315: Provide more information on  preparation of colonic mucosa.

Response: Thank you for your advice. We have added information on preparation of colonic mucosa accordingly.

Answer to authors: It was addressed. Change rpm by g in lines 323.

54) Materials and Methods, line 317: Provide more information on determination of TNF-α, IFN-γ, IL-4, IL-6 and IL-10.

Response: Thank you for your advice. We have added information on determination of cytokines accordingly.

Answer to authors: It was addressed.

55) Materials and Methods, line 320: Provide more information on  preparation of colon tissues.

Response: Thank you for your advice. We have added information on preparation of colon tissues accordingly.

Answer to authors: It was addressed. Change rpm by g in lines 334 and 337.

56) Materials and Methods, line 320: Provide more information on  preparation of Caco-2 cells, how long ere they incubated, etc…

Response: Thank you for your advice. We have added information on preparation of colon tissues accordingly.

Answer to authors: It was addressed.

57) Materials and Methods, line 322: Provide more information how proteins were extracted.

Response: Thank you for your advice. We have added information on preparation of colon tissues accordingly.

Answer to authors: It was addressed.

58) References contained mostly relatively old papers.

Response: Thank you for your advice. We have added some relative newer references in the revised manuscript. 

Answer to authors: It was addressed. 

Author Response

Response to Reviewer 1 Comments

General comments:

1) Polysaccharides are rapidly metabolized by the organism. It is probably unlikely that they may have any effects as anti-inflammatory agents and on improving mucosal barrier function. To fully address the putative positive effects of polysaccharides, diet containing only monosaccharides such as rhamose, glucose and galactose shall have been performed.

1st Response:

Thank you for your suggestions. Exopolysaccharides (EPS) are polymers synthesised by a range of bacterial groups. In general, EPS was not easy to digest which was shown to protect the bacterial cell from environmental stresses, such as human gastric and pancreatic enzymes, bile salts and varying pH [1]. A previous report has improved that EPS produced by Lactococcus lactis ssp. cremoris B40, Lactobacillus sakei 0–1, Streptococcus thermophilus SFi20, and Lactobacillus helveticus Lh59 all retain integrity during gastric transit [2].

For this reason, the undigested EPS can reach the lower intestine and positively impact on the gut microbiome and indirectly modulate the immune system. It was well documented that dendritic cells (DC) sample luminal microbes and microbial components as antigens, such as EPS, resulting in augmentation of natural killer cell activity and modulate inflammatory cytokine expression including interleukins, interferons and tumour necrosis factor [3, 4]. In this study, anti-inflammatory effect of EPS may be related to this mechanism and we will study further next.

Reference:

1.         Ryan, P. M.; Ross, R. P.; Fitzgerald, G. F.; Caplice, N. M.; Stanton, C., Sugar-coated: exopolysaccharide producing lactic acid bacteria for food and human health applications. Food Funct 2015, 6, (3), 679-693.

2.         Ruijssenaars, H. J.; Stingele, F.; Hartmans, S., Biodegradability of food-associated extracellular polysaccharides. Curr Microbiol 2000, 40, (3), 194-199.

3.         Farache, J.; Koren, I.; Milo, I.; Gurevich, I.; Kim, K. W.; Zigmond, E.; Furtado, G. C.; Lira, S. A.; Shakhar, G., Luminal Bacteria Recruit CD103(+) Dendritic Cells into the Intestinal Epithelium to Sample Bacterial Antigens for Presentation. Immunity 2013, 38, (3), 581-595.

4.         Arena, A.; Maugeri, T. L.; Pavone, B.; Iannello, D.; Gugliandolo, C.; Bisignano, G., Antiviral and immunoregulatory effect of a novel exopolysaccharide from a marine thermotolerant Bacillus licheniformis. International immunopharmacology 2006, 6, (1), 8-13.

Answer to authors: This concern was not adequately addressed. There was no experimental evidence that exopolysaccharides remained undigested in mice. It is unclear how such large molecules can interact specifically with tissues and/or endothelial cells. The exopolysaccharides can be digested by microbiome gut. Therefore diet containing rhamose, glucose and galactose shall have been performed to determine possible effects of monosaccharides.

2nd Response: Thank you for your suggestions. As mentioned in the first reply, EPS was not easy to digest by human gastric and intestinal enzymes, but exopolysaccharides can be digested by gut microbiome. However, in this digested progress, gut microbes typically convert these polysaccharides into short-chain fatty acids, which has been shown to have a good effect on alleviating intestinal inflammation, rather than monosaccharides.

2) It is not possible to rule out that other compounds than polysaccharides could have possible beneficial effects. In this respect, sample-to-sample variation was not adequately addressed. The mass spectra was not fully exploited to pint point possible other compounds.

1st Response:

We are sorry for the unclear description. Just as your concern, the purity of EPS is very important for our study. The EPS was isolated and purified as previously described [5]. There were four steps in our study to ensure the purity of EPS. Firstly, after fermentation, trichloroacetic acid (TCA) was added to the culture to a final concentration of 4% (w/v), and the mixture was stirred for 30 min at room temperature. Cells and precipitated proteins were removed by centrifugation (8,000 × g, 4 ◦C, 10 min). Secondly, crude EPS was precipitated from the supernatant by addition of 2 volumes of cold ethanol stored at 4◦C for 24 h. Crude EPS was collected by centrifugation at 8,000 × g for 10 min. Most of the rest proteins, liposoluble and other components were removed. Thirdly, crude EPS solution was dialyzed using a dialysis bag (Mw cut-off 8 kDa to 14 kDa) against distilled water for 48 h at 4 °C to remove the low molecular weight components. Fourthly, before the crude EPS solution was fractionated with an anion exchange chromatography, we tested the sample again for protein, amino acid and polypeptide content to rule out interference from other compounds. We have revised the original manuscript in this part to make it easier for readers to understand.

The average molecular weights of the purified EPS were measured by gel permeation chromatography (GPC) rather than mass spectroscopy. GPC is a commonly used method for determining the average molecular weight of polysaccharides [5, 6]. Standard dextrans (4, 10, 32, 100, 500kDa, Fluka Chemical Co., Buchs, Switzerland) were passed through Waters Ultra-hydrogel 250, 1000 and 2000 (7.8 × 300 mm) columns in series. The column was eluted with 20 mM CH3COONH4 solution at a flow rate of 0.5 mL/min, and the injection volume of sample was 20 μL at an internal temperature of 40 °C. The standard curve equation Log Mw = -0.1741x + 11.505 (R2 = 0.9913), where Mw is the peak molecular weight and x is the retention time (figure 1 below).

Figure 1 Standard curve of dextrans measured by gel permeation chromatography

Thank you for your suggestions. In the revised manuscript, molar ratio of rhamnose, glucose, galactose and mannose was changed to 12.9: 26.0: 60.8: 0.25. We analyzed the three extracted EPS samples. The approximate molar ratio of the sugar monomers was shown in revised figure 1C and the table below. The relative proportion of four different sugar monomers varied slightly between samples. The proportion of mannose was very low (0.25%).

Sample NO.

Rhamnose(%)

Glucose(%)

Galactose(%)

Mannose(%)

1

12.54

25.81

61.40

0.25

2

13.43

26.14

60.20

0.23

3

12.68

25.97

61.08

0.27

Average proportion

12.88

25.97

60.8%

0.25

Standard deviation

0.48

0.16

0.62

0.02

Reference:

5.         Zhang, J.; Cao, Y.; Wang, J.; Guo, X.; Zheng, Y.; Zhao, W.; Mei, X.; Guo, T.; Yang, Z., Physicochemical characteristics and bioactivities of the exopolysaccharide and its sulphated polymer from Streptococcus thermophilus GST-6. Carbohydrate polymers 2016, 146, 368-75.

Answer to authors: It was not adequately addressed. The authors give convincible information on composition and purity of polysaccharides. However, it is not clear how undigested exopolysaccharides can interact with heterogeneous human epithelial colorectal adenocarcinoma cells (Caco-2 cell line). A careful inspection of alternate compounds is needed to fully convince that an intact large polysaccharide can interact with human epithelial cells.

2nd Response: Thank you for your suggestions. Intestinal epithelial cells, including Caco-2 cells, could express some of the molecules of the TLR family, which belongs to the pattern recognition receptor (PRR) family. Polysaccharides are usually considered to be the important ligands for receptor proteins such as TLR-4 and TLR-2. In this study, EPS could interact with intestinal epithelial cells by this way and we will study further next.

3) The molecular mechanisms of polysaccharides and of monosaccharides to trigger anti-inflammatory response or improving mucosal barrier function need to be tackled to fully convince that such molecules could have any biological effects.

1st Response:

Thank you for your suggestions. As discussed above, EPS was not easy to digest, the undigested intact EPS can reach the intestine and positively impact on the gut microbiome and indirectly modulate the immune system. Dendritic cells (DC) sampled EPS as antigens, resulting in immunoregulation of inflammatory cytokine expression including interleukins, interferons and tumour necrosis factor [3, 4]. In this study, anti-inflammatory effect of EPS maybe related to this mechanism and we will study further next.

Reference:

3.         Farache, J.; Koren, I.; Milo, I.; Gurevich, I.; Kim, K. W.; Zigmond, E.; Furtado, G. C.; Lira, S. A.; Shakhar, G., Luminal Bacteria Recruit CD103(+) Dendritic Cells into the Intestinal Epithelium to Sample Bacterial Antigens for Presentation. Immunity 2013, 38, (3), 581-595.

4.         Arena, A.; Maugeri, T. L.; Pavone, B.; Iannello, D.; Gugliandolo, C.; Bisignano, G., Antiviral and immunoregulatory effect of a novel exopolysaccharide from a marine thermotolerant Bacillus licheniformis. International immunopharmacology 2006, 6, (1), 8-13.

Answer to authors: It was not adequately addressed. As the authors pointed out intact polysaccharide shall have an impact on gut microbiome. In this report there were no findings on the effects of intact exopolysaccharide to gut microbiome.

2nd Response: Thank you for your suggestions. The aim of this study was to investigate the alleviating effect of the purified EPS on murine model of colitis, which was not related to relationship between gut microbiome. However, we have proved that EPS-1 had significant impact on gut microbiome of the mice. As a matter of fact, we have analyzed the effect of EPS-1 on gut microbiome by 16srDNA high-throughput sequencing. The related article is being written.

4) The coverage of literature was not critically assessed and some paragraphs lacked appropriate references.

1st Response:

We are sorry for the careless and these mistakes. We have corrected these mistakes in revised manuscript.

Answer to authors: It was not adequately addressed. The relationship between gut microbiome and epithelial cells was not sufficiently explained. A more critical assessment of literature shall be performed especially that there are confusions on the true target of exopolysaccharides and mechanisms of inflammation.

2nd Response: Thank you for your suggestions. The aim of this study was to investigate the alleviating effect of the purified EPS on murine model of colitis, which was not related to relationship between gut microbiome and epithelial cells. For all this, the comments of reviewers are very important for our future research. We suspect that exopolysaccharides could alleviate the inflammation by two ways: (1) Polysaccharides are usually considered to be the important ligands for intestinal epithelial cells receptor proteins such as TLR-4 and TLR-2. EPS could bind to the intestinal epithelial cell receptor and stimulate immune response in the gut lamina propria, which will directly trigger anti-inflammatory response; (2) EPS could improve intestinal flora structure and then increase production of short chain fatty acids in the gut, which has been shown to have a good effect on alleviating intestinal inflammation. We will further confirm these hypotheses in future studies.

5) Ethical concerns for animal uses were not addressed.

1st Response:

 Thank you for your suggestions. Ethical concerns for animal was shown below.

Answer to authors: It was not addressed. What is needed is a statement in “4.4 Animals, diets and experimental 281 procedure” where ethical concerns are addressed, as in the case for all experiments on animals. There is no point to provide a copy of a form that is not included in the manuscript.

 2nd Response: We are sorry for the unclear description. I have added these informations in revised manuscript accordingly.

Minor comments:

10) Abstract: Specify Caco-2

1st Response: Thank you for your suggestions. But as far as I know, the Caco-2 cell line is a continuous line of human epithelial colorectal adenocarcinoma cells. “Caco-2” is not an abbreviated form.

Answer to authors: The authors shall add “continuous line of human epithelial colorectal adenocarcinoma” to specify Caco-2 cells.

2nd Response: Thank you for your advice. I have modified it accordingly.

18) Introduction, line 67: Specify MN-BM-A01 (CGMCC No. 11383).

1st Response: Thank you for your suggestions. However, MN-BM-A01 is the full name of the strain rather than the abbreviation. “CGMCC No. 11383” is the strain preservation number in China general microbiological culture collection center.

Answer to authors: It was not addressed. It shall be stated that MN-BM-A01 (CGMCC No. 11383) is a Streptococcus thermophilus strain isolated from Yogurt Block in Gansu, China.

2nd Response: Thank you for your advice. I have modified it accordingly.

20 ) Results line 86: Chromatography profiles indicated the presence of fucose, and mannose in addition to rhamose, glucose and galactose that were not reported in the text.

1st Response: We are sorry for the unclear description and thanks for your suggestions. We added the proportion of mannose and provided a new chromatography profile (fig 1C) in revised manuscript. EPS-1 was composed of four different sugar monomers including rhamnose, glucose, galactose and mannose. We analyzed the three extracted EPS samples. The approximate molar ratio of the sugar monomers was shown in table below. The relative proportion of four different sugar monomers varied slightly between samples. The proportion of mannose was very low (0.25%).

Sample NO.

Rhamnose(%)

Glucose(%)

Galactose(%)

Mannose(%)

1

12.54

25.81

61.40

0.25

2

13.43

26.14

60.20

0.23

3

12.68

25.97

61.08

0.27

Average proportion

12.88

25.97

60.8%

0.25

Standard deviation

0.48

0.16

0.62

0.02

Answer to authors: It was addressed. However, this table shall be put in the manuscript.

2nd Response: Thank you for your advice. I have added this result in revised manuscript accordingly.

24) Results line 112 and Fig 2E: It is unclear when histological scores were determined and so far there were no findings suggesting an increase of histological scores at day 14.

1st Response: We are sorry for the unclear description. At the end of the experiment (day 14), after sacrificed, histological injury score of each colon was evaluated by morphological criteria as the protocol previously described. The length of colons was measured and cut open longitudinally 1 cm of the distal colon for formalin fixing, paraffin embedding and HE staining. The paraffin sections of colon were graded by two blinded investigators with a range from 0 to 3 as to amount of inflammation (acute and chronic), depth of inflammation and with a range from 0 to 4 as to the amount of crypt damage or regeneration as previously described [7]. Histological scores was the sum of these scores.

Reference:

7.    Dieleman, L. A.; Palmen, M. J.; Akol, H.; Bloemena, E.; Pena, A. S.; Meuwissen, S. G.; Van Rees, E. P., Chronic experimental colitis induced by dextran sulphate sodium (DSS) is characterized by Th1 and Th2 cytokines. Clinical and experimental immunology 1998, 114, (3), 385-91.

Answer to authors: It was addressed. However, I suggest to change the sentence from “Histological scores were increased at day 14 in DSS-treated group compared to the control group” to “At the end of experiment (day 14), histological scores were determined. The histological scores in DSS-treated group increased compared to the control group.” The confusion arises from the wrong perception that there were no changes before day 14, while the authors confirmed there were no measurements performed before that time. Therefore the sentence needs to be clarified.

2nd Response: Thank you for your advice. I have modified this sentence in revised manuscript accordingly.

30) Results, lines 150-165: There was no control on the viability of cells and it is unclear if the changes were related to toxicity of the treatment.

1st Response: We are sorry for the unclear description. In cellular experiments, EPS (2mg/mL) treatment alone was regarded as control to measure the toxicity of EPS.

Answer to authors: It is not addressed since the same experiment without EPS-1 shall have been performed as stated in the text “without or with EPS-1 for 24 hours” (line 157) which contradicts the lack of findings (Fig 5A) on treatment without EPS-1 with or without LPS. Besides it is not clear what is the control.

2nd Response: We are sorry for the unclear description. I have modified this sentence in revised manuscript accordingly.

31) Results, line 156, Figure 5D: Provide quantitative estimation of Western Blot profiles.

1st Response: Thank you for your advice. We have added a new fig.5E to show the quantitative analysis of Western Blot profiles

Answer to authors: It was addressed. However, provide statistical significances in Fig 5E to substantiate significant changes in protein expressions.

2nd Response: Thank you for your advice. I have marked statistical significances in Fig 5E

42) Discussion, lines 213-218: The sentences “In our study, EPS-1 treatment attenuated the release of TNF-α, IFN-γ and IL-6 significantly, suggesting that this effect is the primary cause of anti-colitic activity of EPS. The inhibition effect or pro-inflammatory cytokines was further confirmed in Caco-2 cell lines treated by LPS with or without EPS-1. Results demonstrated that EPS-1 attenuated the LPS-induced inflammatory response of the mucosa and the TER reduction,accompanying significantly alleviated morphological disruption of tight junction protein.” Shall be softened since it is unlikely that polysaccharides could have an anti-inflammatory effect, since they are rapidly metabolized.

1st Response:

Thank you for your suggestions. Exopolysaccharides (EPS) are polymers synthesised by a range of bacterial groups. In general, EPS was not easy to digest which was shown to protect the bacterial cell from environmental stresses, such as human gastric and pancreatic enzymes, bile salts and varying pH [1]. A previous report has improved that EPS produced by Lactococcus lactis ssp. cremoris B40, Lactobacillus sakei 0–1, Streptococcus thermophilus SFi20, and Lactobacillus helveticus Lh59 all retain integrity during gastric transit [2].

For this reason, the undigested EPS can reach the lower intestine and positively impact on the gut microbiome and indirectly modulate the immune system. It was well documented that dendritic cells (DC) sample luminal microbes and microbial components as antigens, such as EPS, resulting in augmentation of natural killer cell activity and modulate inflammatory cytokine expression including interleukins, interferons and tumour necrosis factor [3, 4]. In this study, anti-inflammatory effect of EPS may be related to this mechanism and we will study further next.

Reference:

1.         Ryan, P. M.; Ross, R. P.; Fitzgerald, G. F.; Caplice, N. M.; Stanton, C., Sugar-coated: exopolysaccharide producing lactic acid bacteria for food and human health applications. Food Funct 2015, 6, (3), 679-693.

2.         Ruijssenaars, H. J.; Stingele, F.; Hartmans, S., Biodegradability of food-associated extracellular polysaccharides. Curr Microbiol 2000, 40, (3), 194-199.

3.         Farache, J.; Koren, I.; Milo, I.; Gurevich, I.; Kim, K. W.; Zigmond, E.; Furtado, G. C.; Lira, S. A.; Shakhar, G., Luminal Bacteria Recruit CD103(+) Dendritic Cells into the Intestinal Epithelium to Sample Bacterial Antigens for Presentation. Immunity 2013, 38, (3), 581-595.

4.         Arena, A.; Maugeri, T. L.; Pavone, B.; Iannello, D.; Gugliandolo, C.; Bisignano, G., Antiviral and immunoregulatory effect of a novel exopolysaccharide from a marine thermotolerant Bacillus licheniformis. International immunopharmacology 2006, 6, (1), 8-13.

5.         Zhang, J.; Cao, Y.; Wang, J.; Guo, X.; Zheng, Y.; Zhao, W.; Mei, X.; Guo, T.; Yang, Z., Physicochemical characteristics and bioactivities of the exopolysaccharide and its sulphated polymer from Streptococcus thermophilus GST-6. Carbohydrate polymers 2016, 146, 368-75.

6.         Striegel, A. M.; Timpa, J. D., Molecular Characterization of Polysaccharides Dissolved in Me(2)Nac-Licl by Gel-Permeation Chromatography. Carbohyd Res 1995, 267, (2), 271-290.

7.         Dieleman, L. A.; Palmen, M. J.; Akol, H.; Bloemena, E.; Pena, A. S.; Meuwissen, S. G.; Van Rees, E. P., Chronic experimental colitis induced by dextran sulphate sodium (DSS) is characterized by Th1 and Th2 cytokines. Clinical and experimental immunology 1998, 114, (3), 385-91.

Answer to authors: It was not adequately addressed. The sentence needs to be softened to include possible unidentified compounds that may induce anti-inflammation response. I am not convinced that EPS can trigger itself anti-inflammation response by Caco-2 cells. In my opinion it is unlikely to have interaction of intact EPS to endothelial cells triggering anti-inflammatory response.

2nd Response: Thank you for your suggestions. Extracellular polysaccharides have been shown to have anti-inflammatory effects in several articles. We have listed several references in the first response and in the original manuscript. In my opinion, exopolysaccharides could alleviate the inflammation by two ways: (1) Polysaccharides are usually considered to be the important ligands for intestinal epithelial cells receptor proteins such as TLR-4 and TLR-2. EPS could bind to the intestinal epithelial cell receptor and stimulate immune response in the gut lamina propria, which will directly trigger anti-inflammatory response; (2) EPS could improve intestinal flora structure and then increase production of short chain fatty acids in the gut, which has been shown to have a good effect on alleviating intestinal inflammation. We will further confirm these hypotheses in future studies.

45) Materials and Methods, lines  251, 253, 256, 285, 305, 306 and 326: Specify units in %.

1st Response: We are very sorry for the mistakes and we have corrected the text accordingly.

Answer to authors: It was not adequately addressed:  lines 251, line 253, line 257, 288, 310, 344. Replace w:v by units. Indeed, it is not clear in which units are w:v, it could be g:mL or mg:mL.

2nd Response: We are very sorry for the mistakes and we have corrected the text accordingly.

50) Materials and Methods, lines 279: Provide information on ethical committee.

1st Response: Thank you for your suggestions. Ethical concerns for animal was shown below.

Answer to authors: It was not addressed. What is needed is a statement in “4.4 Animals, diets and experimental 281 procedure” where ethical concerns are addressed, as in the case for all experiments on animals. There is no point to provide a copy of a form that is not included in the manuscript.

2nd Response: We are sorry for the unclear description and we have corrected the text accordingly.

53) Materials and Methods, line 315: Provide more information on  preparation of colonic mucosa.

1st Response: Thank you for your advice. We have added information on preparation of colonic mucosa accordingly.

Answer to authors: It was addressed. Change rpm by g in lines 323.

2nd Response: We are very sorry for the mistakes and we have corrected the text accordingly.

55) Materials and Methods, line 320: Provide more information on  preparation of colon tissues.

1st Response: Thank you for your advice. We have added information on preparation of colon tissues accordingly.

Answer to authors: It was addressed. Change rpm by g in lines 334 and 337.

2nd Response: We are very sorry for the mistakes and we have corrected the text accordingly.